# Hypothesize and Verify: Natural-Language Explanations of Vision Model Errors

## Abstract

LLM- and agent-based assistants now bring non-experts into direct ML work, where they probe model failures by asking the assistant in plain language. When such a classifier misclassifies an image, the non-expert needs a reliable account of *why*. Two obstacles stand in the way. No benchmark scores free-form natural-language explanations of such errors, and existing retrieval-based methods can only return sentences from a fixed error corpus. We close both. NEMO is a task and benchmark of 1,200 misclassified images across ImageNet-R, ObjectNet, and ImageNet-D, each varying along a controlled factor (artistic style, viewpoint, low-level attributes), scored by an LLM-as-a-Judge (LLM Match) protocol that asks whether the explanation describes that factor. SciTX is a generation-based method that emulates the scientific method: retrieve observations, hypothesize candidate causes, verify each with a counterfactual intervention, and retain the hypothesis whose intervention shifts the model's prediction farthest toward the ground-truth class. The shift is captured by our Counterfactual Explanation Impact (CEI), which serves as both SciTX's selection signal and a complementary evaluation metric. On NEMO, SciTX outperforms retrieval-based and MLLM-augmented baselines on both LLM Match and CEI, and 30 AI practitioners rank it first across all five helpfulness dimensions, including factuality, specificity, and actionability.

## 1 Introduction

With the advent of the LLM and Agent era, even non-experts without sufficient machine learning (ML) expertise—such as product managers, domain stakeholders, and newcomers to the ML field—can now easily use ML workflows. For example, users who are new to machine learning can build ML models and debug them using only natural language interfaces within agent-based coding environments. In this changing landscape, providing reliable natural-language explanations of why a model failed is increasingly important to users. Unreliable natural-language explanations can lead to incorrect decisions during subsequent model improvement processes or in stakeholder decision-making.

Although various methods have been developed for debugging models, most do not provide users with a direct natural-language interface. For example, saliency-based methods such as CAM (Selvaraju et al., 2017) and LIME (Ribeiro et al., 2016) visually highlight the image regions that influence a model's prediction, but interpreting and reasoning about the actual cause of failure in natural language remains the user's responsibility. Additionally, Concept Bottleneck Models (Koh et al., 2020) restrict explanations to a predefined, fixed concept vocabulary, which makes it difficult to flexibly describe new failure modes. Meanwhile, there are also approaches that retrieve error explanations from a natural language-based error corpus (Jain et al., 2023; Eyuboglu et al., 2022), but these are limited by the scope of the pre-constructed, fixed error corpus. As a result, if a model failure falls outside the categories covered by the existing corpus, it cannot provide an appropriate explanation. Finally, it is possible to generate explanations by directly querying a multi-modal LLM (MLLM), but this approach largely relies on the MLLM's parametric knowledge, which poses the risk of generating plausible-sounding natural language explanations that are not verified against the model's behavior.

From the perspective of explanation evaluation, prior studies have mainly focused on assessing explanations retrieved from predefined sentence pools or error corpora. For example, LBEE (Csurka et al., 2024) proposes metrics for error-related sentence retrieval, but the explanations are chosen from a predefined sentence set and do not evaluate free-form natural-language explanations. Moreover, these approaches emphasize how well the retrieved explanations identify predefined error categories, rather than assessing whether an explanation describes the cause of each individual error. Therefore, existing approaches have limitations in describing novel failure modes outside the corpus or in evaluating open-ended, natural-language explanations.

In this work, we propose **Sci**entific Method-Inspired **T**extual E**X**planation (SciTX), a generation-based method that produces verified natural-language explanations of vision-classifier errors. For the evaluation of free-form natural language explanation, we construct the evaluation protocol, **N**atural-language **E**xplanations for **MO**del Errors (NEMO), for this task, paired with LLM-as-a-Judge protocol (LLM Match) scores and the Counterfactual Explanation Impact (CEI) metric.

The key challenge for generation-based explanation methods is ensuring that the generated explanations are not just plausible, but actually testable. We interpret an explanation as a hypothesis about the cause of the model's failure. If the hypothesized cause is real, an intervention targeting it should shift the model's prediction toward the ground-truth class, while an explanation naming an irrelevant factor should lead to no significant change. SciTX implements this principle from the perspective of scientific methodology, proceeding in four stages. **(1) Observation.** Given a misclassified image, SciTX extracts contrastive observations by comparing correctly classified samples from the ground-truth and predicted classes. **(2) Hypothesis formation.** Based on these observations, SciTX generates candidate causes for the model's failure as natural-language hypotheses. **(3) Experimentation.** For each candidate hypothesis, SciTX performs a counterfactual intervention. **(4) Conclusion.** SciTX selects the explanation whose intervention shifts the model's prediction most toward the ground-truth class.

Given target classifiers, NEMO pairs 1,200 misclassified images with a controlled failure factor per dataset. We draw 400 images from each of ImageNet-R, ObjectNet, and ImageNet-D, which vary in artistic style, viewpoint, and low-level attributes, respectively. Because free-form natural-language explanations admit no single reference answer, reference-based text metrics such as BLEU or ROUGE are inapplicable. We therefore adopt the LLM-as-a-Judge paradigm (Dunlap et al., 2024; Yu et al., 2024; Liu et al., 2023b), which has become the standard for evaluating free-form generation through its strong correlation with human judgment. Our protocol, LLM Match, scores each explanation by whether it names the controlled factor, yielding a reproducible automated measure. We complement LLM Match with CEI, which measures the change in ground-truth probability after applying the intervention derived from an explanation, providing a causal-effect signal that protocol-based scoring alone cannot capture.

On NEMO, with the Qwen-VL family as the explanation-generating MLLM and three target classifiers (ViT, CLIP, SigLIP), SciTX outperforms every retrieval-based and MLLM-augmented baseline on both LLM Match and CEI. We compute LLM Match with three judges drawn from non-Qwen families, namely GPT, Claude, and Gemini, so that the explanation generator and the evaluator never share a model family. The lead persists when baselines adopt the same multi-candidate generation and CEI-argmax selection as SciTX, indicating that the gains stem from explanation content rather than from matched selection. The method ranking is also stable across two distinct image editors used for counterfactual interventions, and the absolute scores scale with backbone capacity from Qwen2.5-VL-7B to Qwen3-VL-30B. A user study with AI practitioners further ranks SciTX first on every one of the five helpfulness dimensions. In summary, we make the following contributions:

- We introduce NEMO, an evaluation protocol for free-form natural-language explanations of vision-model errors and an automated LLM Match scoring procedure.

- We propose SciTX, a method that selects each explanation by counterfactual validation following the scientific method. Its primary methodological contribution is the hypothesize-and-verify procedure that turns each free-form hypothesis into an executable counterfactual intervention and tests it against the target model; observation retrieval and MLLM generation are supporting components, and Table 2 and Table 5 decompose their individual contributions.

- We introduce CEI, a counterfactual-impact metric that serves both as an evaluation signal and as SciTX's hypothesis-selection criterion.

- On NEMO, SciTX outperforms all retrieval- and MLLM-based baselines across three target classifiers and multiple MLLM backbones, and 30 AI practitioners rank it first on every helpfulness dimension.

## 2 Related Work

**Automatic Model Debugging.** Model debugging (Ribeiro et al., 2016; Koh et al., 2020; Nguyen et al., 2024) is a key step in designing and improving machine learning models. Diagnosing model errors guides model improvement and informs deployment decisions (Kirichenko et al., 2023; Darcet et al., 2023; Hyeon-Woo et al., 2023; Geirhos et al., 2019). Class Activation Map (CAM) (Selvaraju et al., 2017; Zhou et al., 2016; Fernandez, 2020) highlights image regions that drive a prediction, while Local Interpretable Model-agnostic Explanations (LIME) (Ribeiro et al., 2016) and SHapley Additive exPlanations (SHAP) (Lundberg & Lee, 2017; Lundberg et al., 2020) attribute a decision to specific input features. Concept Bottleneck Models (CBMs) (Koh et al., 2020; Yang et al., 2023) instead design the model to predict human-understandable concepts at intermediate layers. Counterfactual explanation methods derive concept importance scores from latent perturbations (Kim et al., 2023) or contrastive saliency maps from optimized pixel perturbations (Wang et al., 2023); there the counterfactual produces the explanation, whereas SciTX uses it to verify a free-form explanation generated first. These methods do not describe the cause of an error in free-form natural language, which is what SciTX targets.

**Language-based Model Debugging.** Natural language is among the most accessible forms of explanation for non-experts (Doshi-Velez & Kim, 2017), and LLM-based interfaces have made it the primary medium through which non-experts interact with machine learning models. This motivates a growing line of work that explains classifier failures in natural language, progressing from retrieving sentences out of a fixed corpus toward generating them with language models, and from group-level summaries toward per-sample explanations.

Prior work (Eyuboglu et al., 2022; Jain et al., 2023; Rezaei et al., 2024; Dunlap et al., 2024; Csurka et al., 2024; Shaham et al., 2024) produces natural-language explanations of classifier failures. Retrieval-based methods (Eyuboglu et al., 2022; Jain et al., 2023; Rezaei et al., 2024; Csurka et al., 2024) score multimodal similarities between misclassified samples and a fixed corpus of error sentences and return the closest match. They require manual curation of that corpus and cannot describe errors absent from it.

Recently, various methods have attempted to use language models to verbally explain the failure modes of classifiers at the slice or dataset level. For example, LADDER (Ghosh et al., 2025) uses a text-only LLM to generate hypothesis sentences about classifier biases, and B2T (Kim et al., 2024) extracts bias-related keywords from caption corpora. However, these approaches primarily focus on group-level failure analysis or provide results in forms such as attributes or keywords, rather than complete natural-language explanations. In contrast, SciTX focuses on the still under-explored sample-level generation setting. Specifically, we use an MLLM conditioned on the misclassified image to generate free-form natural language explanations, rather than retrieved templates or keywords. Each candidate's explanation is then tested through a counterfactual intervention that measures its effect on the model's prediction.

**Large Language Models.** LLMs (Dubey et al., 2024; Achiam et al., 2023; Bai et al., 2023; Abdin et al., 2024; Guo et al., 2025) and multi-modal LLMs (MLLMs) (Liu et al., 2023a; Li et al., 2022; 2023; Touvron et al., 2023; Bai et al., 2025a) have advanced rapidly, with strong reasoning capabilities across diverse tasks (Kojima et al., 2022; Wei et al., 2022). Retrieval-Augmented Generation (RAG) (Lewis et al., 2020; Guu et al., 2020; Gao et al., 2023) grounds their outputs in external knowledge. MLLMs combine a vision encoder (Radford et al., 2021; Zhai et al., 2023) with an LLM to understand vision and text jointly.

**Counterfactual Reasoning.** Counterfactual reasoning (Pearl, 2009) asks how a model would behave when one factor of the input is changed, and has become a standard lens for diagnosing what drives a model's decision. Prior counterfactual XAI work mostly produces counterfactual examples in pixel or feature space and measures how the model's prediction shifts (Goyal et al., 2019; Yan & Wang, 2023; Jeanneret et al.,

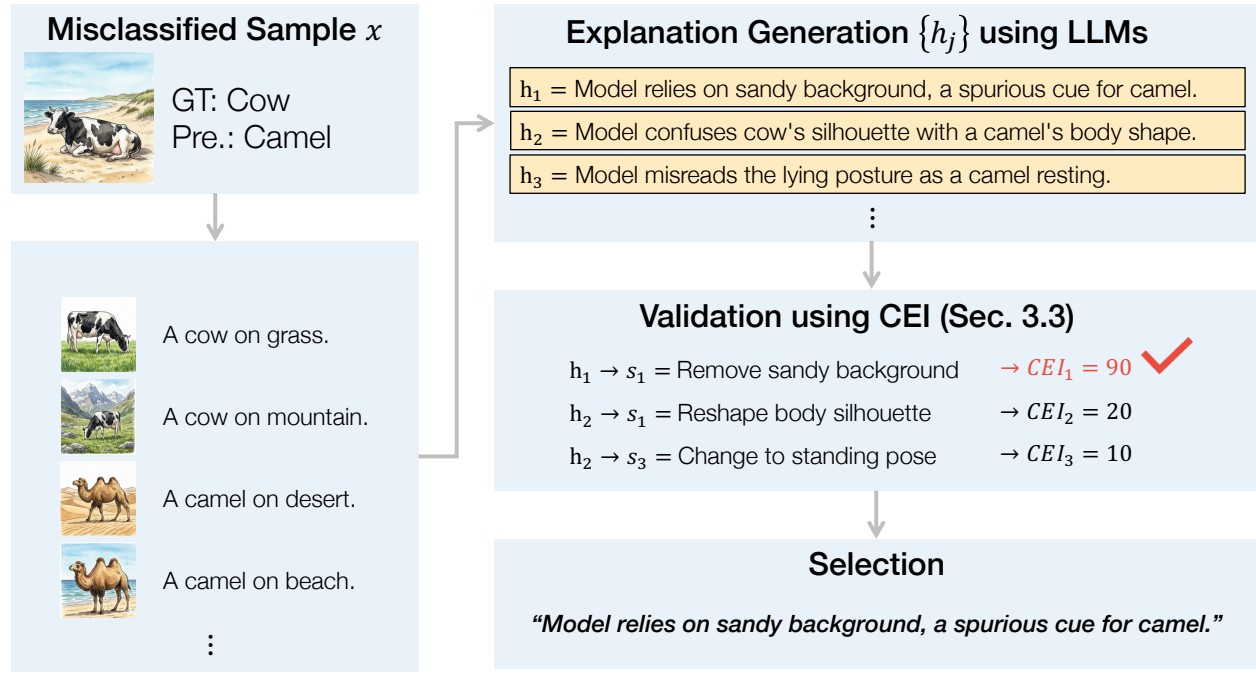

Figure 1: **SciTX.** **(1)** Observation Retrieval. Reference samples from the ground-truth and predicted classes form contrastive context. **(2)** Explanation Generation. An LLM produces candidate hypotheses $\{h_j\}$ from the query and the observations. **(3)** Explanation Validation. Each hypothesis is scored by the CEI of its counterfactual intervention. **(4)** Explanation Selection. The highest-CEI hypothesis is kept as the final explanation. The example is intentionally simple for visual clarity.

2022; Farid et al., 2023; Chiquier et al., 2025), delivering the explanation as a visual diff rather than as a language-level statement of the responsible factor. We adapt this principle to evaluate natural-language explanations. CEI derives an intervention from an explanation through an MLLM and reports the change in the ground-truth probability as a single scalar, providing a quantitative, explanation-conditioned signal. The intervention is applied in image space for standard vision classifiers and in prompt space for VLM-based zero-shot classifiers such as CLIP (Radford et al., 2021) and SigLIP (Zhai et al., 2023), where the zero-shot template is rewritten to incorporate the cue named by the explanation. Recent instruction-following image editors such as FLUX.1-Kontext (Labs et al., 2025), Qwen Image Edit (Wu et al., 2025), and Nano Banana apply targeted edits with high fidelity, making image-space counterfactuals feasible at scale.

## 3 Method

### 3.1 Task Formulation

Let $f : \mathcal{X} \to \mathcal{Y}$ be a discriminative classifier mapping an image $x \in \mathcal{X}$ to a class label, and let $y \in \mathcal{Y}$ denote its ground-truth label. We define the set of misclassified samples as

$$\mathcal{D}_{\mathrm{err}} = \{(x, y) \mid f(x) \neq y\}. \tag{1}$$

Given a sample $(x, y) \in \mathcal{D}_{\mathrm{err}}$ with predicted label $\hat{y} = f(x)$, the **N**atural-language **E**xplanations for **MO**del Errors (NEMO) task is to produce a natural-language explanation

$$e = g(x, y, \hat{y}, f) \in \mathcal{E}, \tag{2}$$

where $g$ is an explanation generator and $\mathcal{E}$ denotes the space of natural-language strings.

We want $e$ to identify the *causal factor* of the misclassification, not just describe the visual content of $x$. We measure this with CEI (Sec. 3.3) and LLM Match (Sec. 3.4).

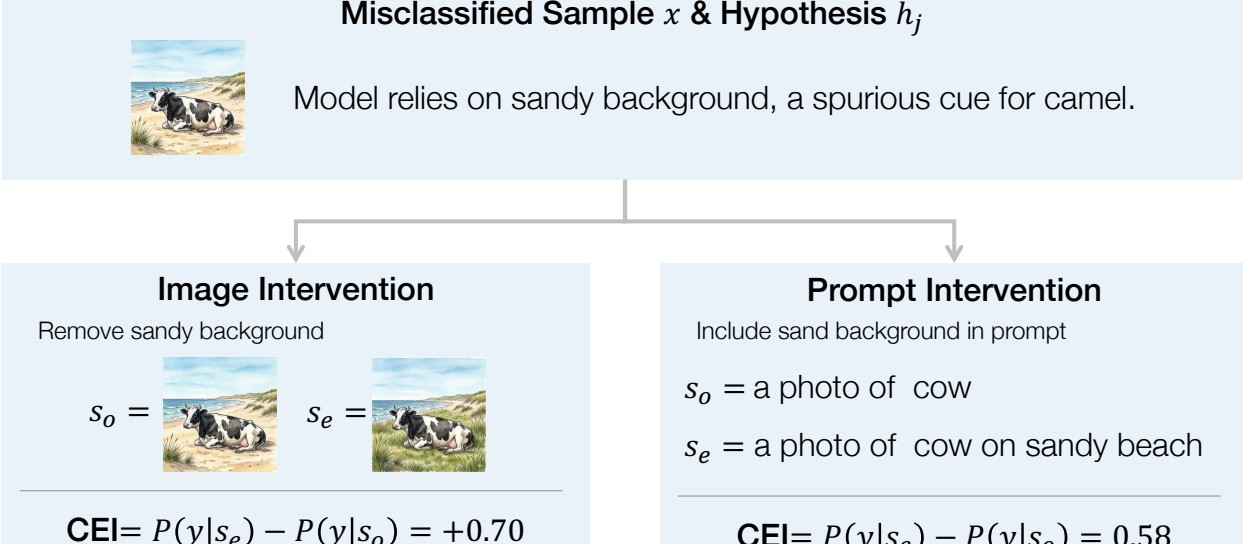

Figure 2: **Counterfactual Explanation Impact (CEI).** CEI measures the change in ground-truth probability after neutralizing the cause described by an explanation, $P(y \mid s_e) - P(y \mid s_o)$. **(Left)** Image Intervention. An image editor transforms the original image $s_o$ into a counterfactual $s_e$ following an MLLM-derived edit. **(Right)** Prompt Intervention. An MLLM rewrites the zero-shot prompt $s_o$ into $s_e$ to incorporate the identified cue.

### 3.2 SciTX

To diagnose the root causes of model errors, we propose **Sci**entific Method-Inspired **T**extual E**X**planation (SciTX), which implements the explanation generator $g$ from Sec. 3.1. SciTX follows the four steps of the scientific method, namely observation, hypothesis, experiment, and conclusion (Fig. 1).

**Observation Retrieval (Survey).** To avoid explanations that rely solely on the MLLM's parametric knowledge, we build a support set $\{o_i\}_{i=1}^{K}$ of correctly classified samples from a reference database $\mathcal{D}$, so that the MLLM can contrast them with the misclassified image $x$. We retrieve observations under three criteria:

- *Correctness*: $f$ classifies $o_i$ correctly, ensuring $o_i$ serves as a reliable reference.

- *Class Relevance*: the label of $o_i$ matches either the ground-truth $y$ or the prediction $\hat{y}$.

- *Semantic Similarity*: $o_i$ is among the $K$ nearest neighbors of $x$ in the visual feature space of $f$.

We denote this process as

$$\{o_i\}_{i=1}^{K} = \text{Retrieve}(x, \mathcal{D}). \tag{3}$$

Like Retrieval-Augmented Generation (RAG; Lewis et al., 2020), this step grounds the MLLM in external evidence so that it can compare the error against valid references. Feeding multiple images to an MLLM can cause *cross-image information leakage* (Park et al., 2025), where the model unintentionally mixes information across them. To avoid this, we represent each observation $o_i$ as a textual caption that the same MLLM produces from the raw image, rather than as the image itself.

**Explanation Generation (Hypothesis).** Using the retrieved observations as context, SciTX generates candidate explanations. Following the broader role-prompting literature (Park et al., 2023; Wang et al., 2024a;b), we assign the MLLM an explicit failure-analyst role so that the hypotheses $\{h_j\}_{j=1}^{M}$ describe possible causes of the misclassification rather than the image content:

$$\{h_j\}_{j=1}^{M} = \text{Hypothesize}\left(x, \{o_i\}_{i=1}^{K}\right). \tag{4}$$

**Validation and Selection (Experiment and Conclusion).** In the final stage, SciTX tests each hypothesis $h_j$ and keeps the most effective one. The MLLM translates $h_j$ into a counterfactual intervention $s_j$, either an image-editing instruction or a modified prompt, and we score $h_j$ with the Counterfactual Explanation Impact (CEI) metric defined in Sec. 3.3. The hypothesis with the largest CEI becomes the final explanation $e$:

$$e = \underset{h_j \in \{h_j\}_{j=1}^M}{\arg\max} \ \text{CEI}(h_j), \quad \text{s.t.} \ \ s_j = \text{Intervention}(h_j). \tag{5}$$

Algorithm 1 summarizes SciTX.

---
**Algorithm 1** SciTX (overview).

---
**Require:** Misclassified image $x$, reference database $\mathcal{D}$, labels $(y, \hat{y})$
**Ensure:** Explanation $e$
1: $\{o_i\}_{i=1}^K \leftarrow \text{Retrieve}(x, \mathcal{D})$                ▷ Observation
2: $\{c_i\}_{i=1}^K \leftarrow \text{Caption}(\{o_i\}_{i=1}^K)$             ▷ via the same MLLM
3: $\{h_j\}_{j=1}^M \leftarrow \text{Hypothesize}(x, \{c_i\}_{i=1}^K)$          ▷ Hypothesis
4: **for** each $h_j$ **do**
5:    $s_j \leftarrow \text{Intervention}(h_j)$              ▷ Experiment
6:    $\text{score}_j \leftarrow \text{CEI}(h_j)$
7: **end for**
8: $e \leftarrow \arg\max_j \text{score}_j$                ▷ Conclusion
9: **return** $e$

---

### 3.3 Counterfactual Explanation Impact (CEI)

We propose the **Counterfactual Explanation Impact (CEI)**, a counterfactual-inference-based metric (Pearl, 2009; Goyal et al., 2019; Yan & Wang, 2023). If $e$ identifies a causal factor of the error, neutralizing that factor should move the model's prediction toward the ground-truth class. CEI measures this shift directly (Fig. 2).

**Definition.** CEI evaluates an explanation through a three-step counterfactual procedure. (i) *Derivation* extracts an intervention instruction from $e$. (ii) *Intervention* applies the instruction to the inference source and constructs a counterfactual scenario. (iii) *Estimation* measures the resulting change in the ground-truth probability. To make the metric reflect $e$, we never reveal the true causal factor of the error to the MLLM; the intervention is derived from $e$ alone.

Formally, let $s_o$ denote the original inference source and $s_e$ its counterfactual counterpart derived from $e$. CEI is defined as

$$\text{CEI}(e) = P(y \mid s_e) - P(y \mid s_o), \tag{6}$$

where $P(y \mid \cdot)$ is the model's probability assigned to the ground-truth label $y$, so a higher CEI provides stronger evidence that $e$ captures a causal factor of the misclassification. We define CEI in two settings, one for standard vision classifiers and one for vision-language models.

**Counterfactual Image Intervention.** For standard discriminative vision models, we intervene directly in the image space (Chiquier et al., 2025; Farid et al., 2023; Jeanneret et al., 2022). As illustrated in Fig. 2 (Left), an MLLM-guided image editor removes the factor described by $e$ from the input image. The original source is the input image $s_o = \text{Img}$, and its counterfactual is the edited image

$$s_e = \mathcal{G}(\text{Img}, \tau), \tag{7}$$

where $\mathcal{G}$ is a subject-driven image-editing model (Labs et al., 2025) and the editing instruction $\tau$ (e.g., *"Remove the sandy background"*) is derived from $e$ by an MLLM.

**Counterfactual Prompt Intervention.** For VLM-based zero-shot classifiers such as CLIP, we instead intervene in the textual prompt space, which modulates the classifier's decision boundary without altering

image pixels. As illustrated in Fig. 2 (Right), an MLLM rewrites the zero-shot prompt to incorporate the cue described by $e$. Let $s_o$ be a standard zero-shot prompt template (e.g., *"a photo of a {class}"*). The counterfactual prompt is

$$s_e = \mathcal{L}(s_o, e), \tag{8}$$

where the MLLM $\mathcal{L}$ rewrites $s_o$ to explicitly incorporate the cue identified by $e$ (e.g., *"a photo of a {class} on a sandy beach"*).

**What CEI establishes.** A positive CEI shows that intervening on the factor described by $e$ moves the model's prediction. This is evidence that the factor is causally relevant to the error, not an identification of the reasoning the model originally used. The intervention can also change more than the described factor; an edit may alter several attributes at once or simply produce an easier image. Appendix C.3 quantifies these concerns, showing that a quality-enhancement edit alone yields near-zero CEI and that an explanation aimed at a mismatched cause recovers only part of the matched effect. We therefore read a high CEI as causal-relevance evidence rather than a mechanistic account of the failure.

**Relation to SciTX.** SciTX uses CEI as its internal hypothesis-selection signal (Sec. 3.2), so CEI does not independently validate SciTX; a high CEI for SciTX partly reflects that SciTX selects for it. We therefore rely on LLM Match (Sec. 3.4) and the human study (Sec. 4.3) as checks that do not favor SciTX by construction, and Table 2 further shows that granting every baseline the same CEI-based selection does not close the gap.

### 3.4 LLM Match

We report two metrics and interpret them jointly. The first is the **LLM Match**, which assesses whether an explanation describes the dataset's controlled factor. The second is CEI (Sec. 3.3), which directly measures the causal effect of an explanation. Neither metric is conclusive on its own. LLM Match can be satisfied by naming the controlled factor without any sample-specific content (Appendix C.1), and CEI is also SciTX's hypothesis-selection signal, so we draw cross-method conclusions from the two metrics together.

NEMO comprises misclassified samples from three robustness benchmarks, namely ImageNet-R (Hendrycks et al., 2021), ObjectNet (Barbu et al., 2019), and ImageNet-D (Zhang et al., 2024), each constructed to vary along a specific factor. ImageNet-R varies artistic style and rendition; ObjectNet varies viewpoint, rotation, and background; ImageNet-D varies background, texture, and material via diffusion generation. We sample 400 misclassified images from each, yielding 1,200 in total. By construction, the controlled factor is the dominant driver of misclassification on each benchmark, so a correct explanation should describe it.

We accordingly define a dataset-specific factor criterion for each benchmark and assess whether the explanation describes it. Given an explanation and the dataset-specific criterion, an LLM-as-a-Judge assigns one of three scores, namely 1.0 (full description), 0.5 (partial description), or 0.0 (no description). To reduce single-judge bias, we average scores from three LLM judges (Sec. 4.1.1). Appendix C.5 reports inter-judge agreement and a calibration of LLM Match against human annotations from our user study.

## 4 Experiments

### 4.1 Setting

#### 4.1.1 Models

**Target Models.** We employ two VLMs, CLIP (Radford et al., 2021) and SigLIP (Zhai et al., 2023), sourced from Hugging Face, alongside a standard ViT fine-tuned on ImageNet (Steiner et al., 2022; Dosovitskiy et al., 2021) sourced from Timm library (Wightman, 2019).[1]

**MLLM Backbone.** For explanation generation, we adopt the Qwen-VL series (Bai et al., 2025b;a) as a representative high-performing open-weight MLLM family, which keeps the pipeline reproducible without proprietary API access.

---

[1]CLIP: openai/clip-vit-base-patch32, SigLIP: google/siglip-base-patch16-224, ViT: vit_base_patch16_224.augreg2_in21k_ft_in1k.

Table 1: **Quantitative results across three datasets.** SciTX outperforms all baselines on both metrics. **Bold** marks the best per column.

(a) LLM Match averaged over GPT, Claude, and Gemini

| MLLM | ImageNet-R | | | ObjectNet | | | ImageNet-D | | |
|---|---|---|---|---|---|---|---|---|---|
| | ViT | CLIP | SigLIP | ViT | CLIP | SigLIP | ViT | CLIP | SigLIP |
| Retrieval | 0.277 | 0.270 | 0.268 | 0.499 | 0.492 | 0.493 | 0.142 | 0.147 | 0.145 |
| CamMLLM | 0.111 | 0.066 | 0.044 | 0.640 | 0.632 | 0.603 | 0.498 | 0.522 | 0.467 |
| ChangeMLLM | 0.279 | 0.309 | 0.291 | 0.761 | 0.744 | 0.758 | 0.390 | 0.373 | 0.358 |
| **SciTX** | **0.558** | **0.632** | **0.594** | **0.772** | **0.781** | **0.797** | **0.562** | **0.619** | **0.644** |

(b) CEI

| MLLM | ImageNet-R | | | ObjectNet | | | ImageNet-D | | |
|---|---|---|---|---|---|---|---|---|---|
| | ViT | CLIP | SigLIP | ViT | CLIP | SigLIP | ViT | CLIP | SigLIP |
| Retrieval | 11.29 | 11.22 | 1.81 | 16.14 | 7.86 | 0.82 | 6.35 | 3.31 | 0.70 |
| CamMLLM | 13.26 | 3.17 | -0.02 | 15.25 | 7.67 | 0.26 | 7.05 | 5.55 | 0.78 |
| ChangeMLLM | 13.39 | 9.59 | 0.76 | 19.02 | 11.94 | 1.84 | 10.88 | 11.92 | 7.43 |
| **SciTX** | **13.63** | **17.52** | **2.37** | **19.06** | **21.29** | **5.57** | **11.97** | **19.27** | **16.94** |

**Image Editor.** For counterfactual image intervention (Sec. 3.3), we use FLUX.1-Kontext-dev (Labs et al., 2025), an instruction-following image editor.

**Evaluation Judges.** We distinguish reported metrics from SciTX's internal selection signal. For reporting, both LLM Match and CEI are averaged over three non-Qwen judges, namely `gpt-5-mini-2025-08-07`, `claude-haiku-4-5`, and `gemini-3-flash-preview`, so the explanation generator and the evaluator never share a model family. For hypothesis selection inside SciTX (Sec. 3.2), CEI is computed with the same Qwen-VL backbone used for explanation generation.

### 4.1.2 Baselines

**ErrorRetrieval** (Jain et al., 2023; Csurka et al., 2024) retrieves a textual description of the failure from a predefined error corpus, given the misclassified image. Since no prior method tackles per-sample natural-language explanation of classifier errors (Ghosh et al., 2025; Kim et al., 2024), we construct two MLLM-based baselines (CamMLLM, ChangeMLLM). **CamMLLM** feeds Grad-CAM (Selvaraju et al., 2017) saliency maps to an MLLM, which then verbalizes the regions that drove the misclassification. **ChangeMLLM** follows change captioning (Park et al., 2019), asking an MLLM to describe the differences between the misclassified image and a reference image from the ground-truth or predicted class, and then to infer the error cause from those differences.

### 4.2 Results

### 4.2.1 Effectiveness of Our Method

Table 1 reports performance across ImageNet-R, ObjectNet, and ImageNet-D under both metrics. *SciTX outperforms baselines on both metrics across all three datasets.* The baselines instead show dataset-specific weaknesses. Error Retrieval performs comparatively better on ImageNet-R, where its predefined corpus covers rendition-style errors, but collapses on ObjectNet and ImageNet-D, where the corpus does not. Appendix C.2 tests this coverage effect directly by varying the share of relevant sentences in the corpus. ChangeMLLM and CamMLLM also trade ranks across the three datasets. SciTX remains consistently strong, indicating robustness to varying error factors.

### 4.2.2 Fair Comparison and Generalization of Validation Module

SciTX uses CEI both as its hypothesis-selection signal and as one of our reported metrics, so its CEI lead in Table 1 could in principle be explained by selection alone. To rule this out, we augment each baseline

Table 2: **Fair comparison under matched candidate selection (ImageNet-D / CLIP).** Each baseline is augmented with the same multi-candidate generation and CEI-argmax selection used by SciTX. Deltas vs. (w/o) are shown next to (w/) values. Green marks gains and red marks regressions.

| Method | w/o Validation | | w/ Validation | |
|---|---|---|---|---|
| | LLM Match | CEI | LLM Match | CEI |
| Retrieval | 0.147 | 3.31 | 0.150 (+0.003) | 4.72 (+1.41) |
| CamMLLM | 0.522 | 5.55 | 0.513 (-0.009) | 6.26 (+0.71) |
| ChangeMLLM | 0.373 | 11.92 | 0.532 (+0.159) | 12.00 (+0.08) |
| **SciTX** | **0.548** | **12.01** | **0.619** (+0.071) | **19.27** (+7.26) |

Table 3: **Robustness to image intervention model on ImageNet-D / ViT.** The method ranking is preserved across two distinct image editors (FLUX.1-Kontext-dev and Qwen Image Edit).

| Method | CEI | |
|---|---|---|
| | Kontext | Qwen Image Edit |
| Retrieval | $6.351^{(4)}$ | $11.542^{(4)}$ |
| CamMLLM | $7.050^{(3)}$ | $14.290^{(3)}$ |
| ChangeMLLM | $10.879^{(2)}$ | $15.115^{(2)}$ |
| SciTX | $11.967^{(1)}$ | $20.397^{(1)}$ |

Table 4: **Effect of MLLM backbone on ImageNet-R / CLIP.** MLLMs with higher reasoning capability yield higher LLM Match and CEI. **Bold** marks the best per column.

| Method | Qwen2.5-VL-7B | | Qwen3-VL-8B | | Qwen3-VL-30B | |
|---|---|---|---|---|---|---|
| | LLM Match | CEI | LLM Match | CEI | LLM Match | CEI |
| CamMLLM | 0.066 | 3.17 | 0.216 | 4.94 | 0.213 | 7.31 |
| ChangeMLLM | 0.309 | 9.59 | 0.403 | 12.24 | 0.483 | 18.60 |
| **SciTX** | **0.632** | **17.52** | **0.725** | **20.76** | **0.816** | **25.09** |

Table 5: Ablation on observation retrieval strategies (ImageNet-D, CEI). Adding class relevance to similarity-based retrieval consistently improves CEI across all target classifiers. **Bold** marks the best per column.

| Observation | ImageNet-D | | |
|---|---|---|---|
| | ViT | CLIP | SigLIP |
| Correctness + Semantic Similarity | 10.31 | 19.05 | 16.84 |
| + Class Relevance | **11.97** | **19.27** | **16.94** |

with the same multi-candidate generation and CEI-argmax selection used by SciTX on ImageNet-D / CLIP. Table 2 shows that CEI improves for all three baselines and LLM Match for two, yet SciTX remains higher. The lead is therefore not produced by matched selection. Even without validation, SciTX reaches a LLM Match of 0.548, above every baseline in both columns, so its advantage originates in the stages preceding verification, namely contrastive observation retrieval and hypothesis generation, while verification adds a further +0.071. The same module also helps the baselines, suggesting that causal-impact selection is broadly useful for explanation generation, not just for SciTX.

### 4.2.3 Robustness to Image Intervention Model

To verify that the CEI ranking reflects explanation quality rather than depending on a particular image editor, we re-evaluate on ImageNet-D / ViT with FLUX.1-Kontext-dev (Labs et al., 2025) and Qwen Image Edit (Wu et al., 2025). Table 3 preserves the method ranking across both editors, indicating that SciTX's explanations specify more effective counterfactual interventions than the baselines. Absolute CEI values differ across editors, since the editor acts as the measuring instrument; we therefore compare absolute values only within a fixed setup. Appendix C.3 further isolates the editor's own contribution to CEI with an enhancement-only control and a shuffled-explanation control.

### 4.2.4 Effect of MLLM Backbone

Table 4 reports performance on ImageNet-R / CLIP across three MLLM backbones (Qwen2.5-VL-7B, Qwen3-VL-8B, Qwen3-VL-30B). Both SciTX and the baselines improve consistently on both LLM Match and CEI as the backbone grows stronger, indicating that stronger MLLM reasoning translates directly into better explanations of model errors.

### 4.2.5 Retrieved Observations

Fig. 3 shows the observations $\{o_i\}$ retrieved by SciTX for the ground-truth and predicted classes, which serve as comparative visual cues (woven texture vs. metal bars) for the subsequent explanation. We further ablate the retrieval strategy. Replacing class-aware retrieval with pure visual similarity drops CEI across all

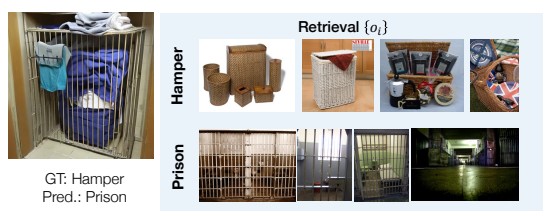

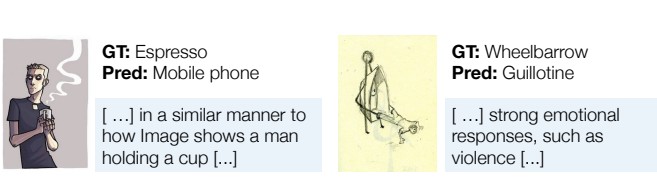

Figure 3: Observations retrieved by SciTX. Reference images for the ground-truth (Hamper) and predicted (Prison) classes that provide comparative visual cues for the explanation.

Figure 4: Beyond pixel-level similarity: (left) an espresso illustration confused with a mobile phone; (right) a stylized wheelbarrow drawing confused with a guillotine.

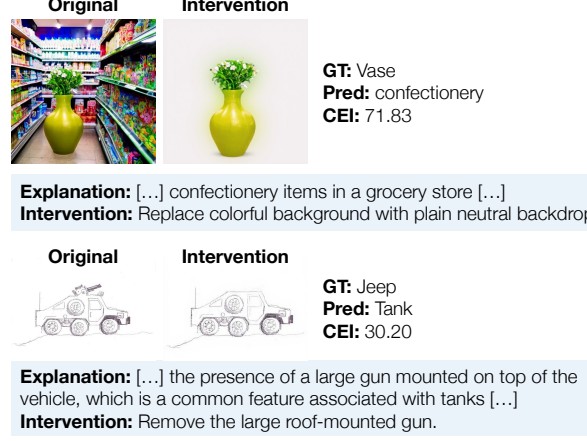

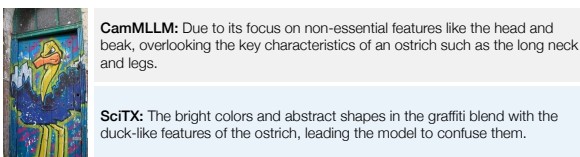

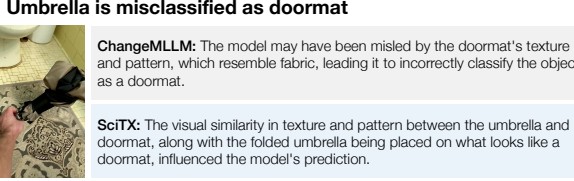

Figure 5: SciTX's explanations translated into counterfactual interventions. CEI measures the change in the ground-truth probability after applying each intervention.

Figure 6: Generated textual explanations. CamMLLM describes the localization information; ChangeMLLM and ours describe the differences or commonalities between the misclassified sample and observations.

target classifiers (Table 5). The gain is largest on ViT, suggesting that grounding observations in both the ground-truth and predicted classes is helpful, especially for non-VL classifiers.

### 4.2.6 Qualitative Analysis

**Generated Explanations.** In the cases shown in Fig. 6, SciTX's explanations refer to multiple visual factors from the retrieved observations, while the baselines describe only one. For the ostrich graffiti, CamMLLM only highlights the head/beak region, whereas SciTX attributes the error to graffiti colors blending with duck-like features. For the umbrella-on-doormat, ChangeMLLM cites only texture similarity, whereas SciTX further invokes the spatial context of a folded umbrella on a doormat-like surface.

**Counterfactual Intervention Examples.** Fig. 5 shows how SciTX's explanations translate into counterfactual interventions. A vase misclassified as confectionery is explained by its colorful grocery-store background, and the intervention replaces the background with a neutral backdrop. A jeep misclassified as a tank is explained by its roof-mounted gun, and the intervention removes it.

**Pose- and Affect-Aware Reasoning.** We identify representative qualitative cases where SciTX invokes high-level cues beyond pixel-level similarity (Fig. 4). For example, an espresso illustration is misclassified as a mobile phone because the figure's holding pose mirrors that of phone-using subjects, and a stylized wheelbarrow drawing is misclassified as a guillotine because both evoke a violent emotional tone.

Table 7: **Computational cost on a single NVIDIA RTX A6000 (ViT / FLUX setting).** (a) Per-sample wall-clock cost of each explainer. (b) SciTX per-sample cost by stage.

(a) Per-sample wall-clock cost

| Method | Time / sample | Main cost |
|---|---|---|
| Retrieval | 0.02s | embedding search only |
| CamMLLM | 1.82s | 1 VLM call |
| ChangeMLLM | 3.35s | 1 VLM call |
| SciTX | 158.5s | $M{=}5$ FLUX edits |

(b) SciTX cost by stage

| Stage | Calls $\times$ time | Time | Share |
|---|---|---|---|
| Contrastive caption generation | $10 \times 1.09$s | 10.9s | 6.9% |
| Hypothesis generation ($M{=}5$) | $5 \times 2.37$s | 11.9s | 7.5% |
| Edit-instruction compilation | $5 \times 1.74$s | 8.7s | 5.5% |
| FLUX editing (verification) | $5 \times 25.4$s | 127.0s | 80.1% |

### 4.2.7 Application: Global Explanation

SciTX can also be used to characterize a model's failure modes at the dataset level. We use an MLLM to aggregate all sample-wise explanations produced by SciTX into a single concise global explanation. For CLIP on ImageNet-R, the resulting summary identifies the dominant failure mode as follows:

> *"[. . . ] struggles with domain shifts, frequently misclassifying stylized representations—such as origami, tattoos, and cartoons—based on **their artistic style rather than their actual structural characteristics**."*

This description is consistent with ImageNet-R being a rendition benchmark, suggesting that the per-sample explanations carry a distribution-level signal that emerges naturally under aggregation.

As a single-condition probe on CLIP / ImageNet-R with Qwen2.5-VL-7B, we plug it back into evaluation as the explanation supplied to the CEI scorer. As reported in Table 6, CEI improves from 17.52 with per-sample explanations to 19.42 with the single global explanation, indicating that the aggregated description preserves the per-sample causal signal under this single condition. We view this as preliminary evidence that SciTX can also serve as a dataset-level failure-mode summarizer.

Table 6: CEI of SciTX on CLIP / ImageNet-R.

| | |
|---|---|
| Local | 17.52 |
| + Global | 19.42 (+1.9) |

### 4.2.8 Computational Cost

We measure per-sample wall-clock cost for every explainer on a single NVIDIA RTX A6000 in the ViT / FLUX image-editing setting, the most expensive configuration. As shown in Table 7, SciTX is by far the most expensive explainer at 158.5s per sample, and this runtime is its main practical limitation. The cost concentrates in one place; the $M{=}5$ FLUX edits take 80% of the total (Table 7b), so the cost grows roughly linearly with $M$ and the edits can run in parallel across GPUs. $M$ also trades cost for accuracy. At $M{=}1$ the pipeline stops after the first hypothesis, taking 13.3s per sample, a $12\times$ reduction, while LLM Match drops by only 0.071 and still exceeds every baseline (Table 2). In the CLIP and SigLIP settings the intervention is a prompt rewrite, so no editor is called and the diffusion cost disappears. Finally, SciTX is an offline diagnostic tool run once over a fixed error set, so the cost is a one-time analysis expense rather than serving latency.

### 4.3 Human Evaluation

**Study Design.** We evaluate on 30 misclassified samples (10 from each of ImageNet-R, ObjectNet, and ImageNet-D). To ensure that all methods produce non-trivial explanations, we restrict to samples whose harmonic mean of CEI across methods is relatively high. We use the harmonic mean to avoid selecting in favor of any single method; it is dominated by the lowest-scoring method, so a sample passes only when every method, including the weakest baseline, produces a non-trivial explanation. This criterion also means the study samples lean toward causes that are amenable to intervention; Appendix C.6 discusses causes outside this set, such as 3D viewpoint changes. Each sample is paired with four candidate explanations, and 30 AI practitioners recruited via Prolific[2] rank them along five dimensions (Overall Helpfulness, Factuality,

---

[2]https://www.prolific.com/. Participants were compensated at £10/h; the average completion time was 57 minutes.

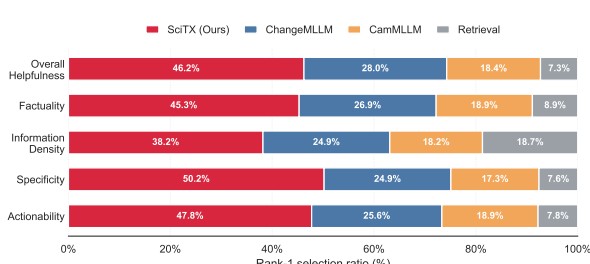

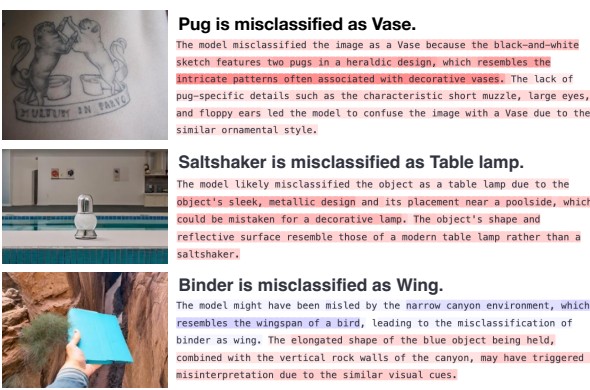

Figure 7: Rank-1 selection ratio of different helpfulness dimensions of human evaluation. We observe that AI practitioners rank our method first more often than the others.

Figure 8: Human highlight result. Red means that humans perceive the explanation is good; blue means that humans perceive the explanation is bad. We average the highlight results of participants and visualize using a color spectrum.

Information Density, Specificity, and Actionability) using a forced ranking without ties. We recruit AI practitioners rather than non-expert end users, since ranking error explanations on dimensions such as factuality and actionability requires enough ML background to judge them. The study therefore measures helpfulness as judged by practitioners, and evaluation with the non-expert users of our motivation remains future work. See Appendix D for the evaluation interface and dimension definitions.

**Results.** As shown in Fig. 7, SciTX consistently achieves the highest rank-1 selection ratio across all five evaluation dimensions (see detailed results in Appendix Fig. 15). Qualitatively, users highlight good (red, +1.0) or bad (blue, −1.0) parts of SciTX's explanations, averaged across participants and visualized on a color spectrum (Fig. 8). Users mark concrete visual cues such as "heraldic design" and "sleek, metallic design" positively, but mark logical leaps such as "narrow canyon ... resembles the wingspan of a bird" negatively. These rankings measure perceived helpfulness rather than correctness. A preference win shows that practitioners find SciTX's explanations more convincing and easier to act on, not that the explanations are causally accurate; a convincing but wrong explanation could also rank highly. Whether an explanation describes the controlled factor and whether its intervention shifts the prediction are assessed separately by LLM Match and CEI. Appendix C.6 analyzes the cases where CEI and the human judgments disagree.

### 4.4 Failure Cases

We observe two failure modes in SciTX's outputs (Fig. 9). First, SciTX may build the explanation on top of an object that the MLLM has misidentified, as in the padlock case where the explanation discusses a basket–hamper handle resemblance without ever referring to the padlock. Second, it may rely on shared scene context rather than discriminative object features, as in the chair-vs-bench case where the confusion is attributed to a common brick-wall background.

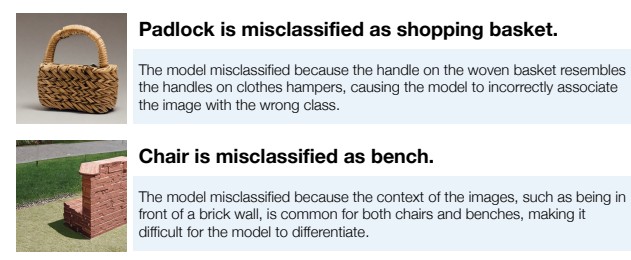

Figure 9: Failure cases.

## 5 Conclusion

We introduce NEMO, a task and benchmark for natural-language explanations of vision-model errors. We further propose SciTX, a generation-based method that follows a four-stage scientific-method-inspired pipeline, verifying each candidate explanation with a counterfactual intervention before selecting one. On NEMO,

SciTX outperforms every retrieval-based and MLLM-augmented baseline on both metrics. A user study with 30 AI practitioners further ranks SciTX first on all five helpfulness dimensions. Future work includes extending the pipeline to broader debugging settings such as generative-model failures and multi-modal classifiers.

**Broader Impact Statement**

As LLM-based interfaces become standard for interacting with deployed models, explanations that describe plausible failure causes can help practitioners and non-expert users decide whether to trust, escalate, or retrain a model. As with any MLLM-generated text, the explanations are best treated as a starting point for inspection rather than a verified diagnosis, and our pipeline is intended to operate within the review-and-iterate workflow that already governs model deployment. Beyond this, we do not foresee additional societal consequences that warrant separate discussion.

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

**Contents**

# A    Prompts

## A.1    SciTX

The prompt of SciTX is given below.

```
You are an expert in visual reasoning and model error analysis.

The first image has the true label **{true_label_1}**, but the model incorrectly predicted it as
**{prediction}**.

***Task***
Your goal is to identify visual differences between these sets of images and analyze the likely
reason for the models misclassification.

***Steps***
1. Examine all provided sets (A, B, and/or C).
   - If a set is missing, use the remaining sets for comparison or inference.
2. Identify visual or semantic differences relevant to the misclassification.
3. Consider whether factors like background, object texture, pose, or context influenced the error.
4. Formulate a concise, evidence-based explanation.

***Evidence Sets***
- **Set A** (True Label: {true_label_1})
  {set_a_captions}

- **Set B** (Predicted Label: {prediction})
  {set_b_captions}

- **Set C** (Random Examples)
  {set_c_captions}

***Output Format***
Write your reasoning **strictly** in JSON format as shown below.

```json
{
   "cause_of_error": "A concrete, evidence-based conclusion directly explaining why the model
   predicted '{prediction}' instead of '{true_label_1}'. Mention the most influential visual cue"
}
```

Do not include any text outside of the JSON object in your output.
```

## A.2    Baseline Prompts

For completeness, we list the generation prompts used by the MLLM-augmented baselines. ErrorRetrieval is non-generative and does not use a prompt.

**CamMLLM.**

```
You are an expert visual reasoning assistant.

Given the Grad-CAM visualization, the model prediction, and the correct label, identify the main
cause of the misclassification.

Your task is to analyze why the model made the wrong prediction.

Follow these reasoning steps:
1. Describe what regions the Grad-CAM highlights (what the model focused on).
2. Assess whether these regions are relevant or misleading for the true label.
3. Identify any important visual cues for the true label that were ignored.

Finally, summarize your reasoning in **strict JSON format** as follows:
```json
{
  "cause_of_error": "concise and precise sentences explaining the most probable cause of
  misclassification"
}
```

Inputs:
Prediction: {prediction_cls_name}
True label: {true_cls_name}

Do not include any text outside of the JSON object in your output.
```

**ChangeMLLM.**

```
You are an expert visual reasoning assistant.

The first image is {true_cls_name}, but the model predicted it as {prediction_cls_name} (incorrect).

The second image is {src_cls_name}, and the model predicted it correctly.

Your task is to identify the differences between these images and analyze why the model made the
wrong prediction.

Follow these steps:
1. Describe key visual differences between the two images.
2. Suggest what kind of visual or contextual cue the model failed to capture.

Finally, summarize your reasoning in **strict JSON format** as follows:
```json
{
  "visual_differences": "Describe the main differences between the two images",
  "cause_of_error": "1-2 concise and precise sentences explaining the most probable cause of
  misclassification"
}
```

Do not include any text outside of the JSON object in your output.
```

### A.3 Evaluation

For counterfactual image intervention:

```
An image with the true label **{true_label_1}** was incorrectly predicted as **{prediction}**.

**Context for Error:**
- **Hypothesis for Misclassification:** {hypothesis}

Your task is to write a concise editing prompt that would modify the image to correct this error.
The prompt should describe a minimal visual change that helps a generative model fix the issue (e.g
., "Add a hat to the cat").

**Guidelines:**
1. **Address the Hypothesis:** The edit must directly resolve the hypothesized cause of the
misclassification.
2. **Be Specific & Semantic:** The edit must be a *content-based* visual change (e.g., "Add a stem
to the apple," "Change the shirt color to blue"). Do NOT suggest generic, non-specific edits like "
make {true_label_1} larger," "sharpen the image," or "increase contrast."
3. **Avoid the Class:** Do NOT reference or imply the incorrect label **{prediction}** in your
solution.
4. **Be Concise:** Output only one short, imperative command (ideally under 10 words).

**Output Format:**
Write your reasoning strictly in JSON format as shown below.

```json
{
   "reasoning": "Briefly explain why this new prompt is better. The reasoning should be grounded in
    Hypothesis for Misclassification.",
   "solution": "Short, concise, imperative editing prompt" # 1 sentence and lower than 10 words
}
```
Do not include any text outside of the JSON object in your output.
```

For counterfactual zero-shot prompt intervention:

```
An image with the true label **{true_label_1}** was incorrectly predicted as **{prediction}**.

**Diagnostic Context:**
1. **Specific Hypothesis:** {hypothesis}
   *(The specific visual reason this image was misclassified.)*
2. **Original Failed Prompt:** "{original_prompt}"

**Task:**
Generate a **concise, CLIP-friendly zero-shot prompt** to correct this error.
Unlike human descriptions, CLIP prefers simple, direct captions. Avoid complex sentences or lists
of features.

**Strict Guidelines for CLIP:**
1. **Keep it Short:** Use fewer than 15 words.
2. **Focus on the Scene:** Instead of listing body parts (e.g., "showing legs, neck, beak"),
describe the overall scene or action (e.g., "standing in a grassy field").
3. **No "Checklists":** Do NOT use phrases like "clearly showing," "distinctive features," or "
characterized by."
4. **No Negatives:** Do NOT use words like "not," "no," or "unlike." (e.g., instead of "not a close
-up," say "a full body shot").
5. **Address the Hypothesis:** Subtlely include the missing visual cue identified in the Hypothesis
, but blend it naturally into the caption.

**Output Format:**
Write your reasoning strictly in JSON format:
```

```json
{
    "reasoning": "Briefly explain how this simplified prompt targets the hypothesis without over-
    complicating.",
    "solution": "a photo of [adjective] {true_label_1} [context/action]"
}
```

## B  Experimental Details

### B.1  Datasets

Dataset-level scope (controlled factor, 400-per-dataset sampling, 1,200 total) is described in Sec. 3.4. Below we record source and reproducibility detail.

**ImageNet-R** ImageNet-R (Hendrycks et al., 2021) covers 200 classes across 30,000 rendition images of ImageNet classes.

**ImageNet-D** ImageNet-D (Zhang et al., 2024) is released under the MIT License. Its images are generated by diffusion models.

**ObjectNet** ObjectNet (Barbu et al., 2019) contains images captured in challenging conditions, *e.g.*, varied backgrounds, rotations, and imaging viewpoints.

**Observation Source** For retrieval, the observation source $\mathcal{D}$ pools the un-sampled examples from these three benchmarks together with a 100-class subset of ImageNet (Deng et al., 2009).

### B.2  Models

**Multimodal Large Language Models** For explanation generation, we employ the Qwen-VL series (Bai et al., 2025b;a). To evaluate the generated outputs, we average scores from three LLM judges, namely `gpt-5-mini-2025-08-07`, `claude-haiku-4-5`, and `gemini-3-flash-preview`. For all models, the top-p and temperature parameters are set to 0 to ensure deterministic results.

**Image Editing Model** We utilize two instruction-following image editing models for our counterfactual interventions, namely FLUX.1-Kontext-dev (Labs et al., 2025) and Qwen Image Edit (Wu et al., 2025).

**Computational Resources** All experiments are conducted on a single NVIDIA RTX A6000 GPU. This setup is sufficient for our study as we use efficient inference APIs to minimize computational overhead.

## C  Experimental Results

### C.1  Constant Baseline for LLM Match

Since LLM Match asks whether an explanation describes the dataset's controlled factor, and that factor is shared by every sample in a benchmark, a fixed dataset-keyed template can name the factor without referencing the image. We construct such a constant baseline, one template sentence per dataset that identifies the controlled factor, and evaluate it on ImageNet-R / CLIP with the same three judges.

Table 8 shows a clear dissociation. The constant baseline achieves the highest LLM Match of all methods (0.809, compared to SciTX's 0.632) but falls to nearly the lowest CEI (4.04, compared to SciTX's 17.52). Because the template carries no sample-specific content, the intervention it induces is generic and fails to address the cause of each individual failure. Naming a dataset's controlled factor is therefore not sufficient to explain why a particular image was misclassified. This is why NEMO reports LLM Match and CEI together rather than in isolation (Sec. 3.4).

Table 8: **Constant baseline on ImageNet-R / CLIP.** A fixed dataset-keyed template that names the controlled factor without referencing the image attains the highest LLM Match but nearly the lowest CEI. **Bold** marks the best per column.

| Method | LLM Match | CEI |
|---|---|---|
| Constant baseline | **0.809** | 4.04 |
| Retrieval | 0.270 | 11.22 |
| CamMLLM | 0.066 | 3.17 |
| ChangeMLLM | 0.309 | 9.59 |
| SciTX | 0.632 | **17.52** |

Table 9: **Effect of corpus coverage on retrieval (ImageNet-R / CLIP).** Percentages denote the share of style sentences in the retrieval corpus (14.0% is the original corpus). Style pick is how often the retriever selects a style sentence. Retrieval tracks coverage on every axis, while SciTX does not use the corpus and is unaffected. All values are re-evaluated in this experiment's run and may differ slightly from Table 1.

| Method | LLM Match | CEI | Style pick |
|---|---|---|---|
| Retrieval (9.3%) | 0.242 | 11.23 | 39.5% |
| Retrieval (14.0%) | 0.270 | 11.54 | 43.8% |
| Retrieval (42.7%) | 0.432 | 27.58 | 77.2% |
| Retrieval (57.0%) | 0.465 | 32.71 | 84.5% |
| SciTX | 0.632 | 17.60 | – |

## C.2 Effect of Corpus Coverage on Retrieval

The retrieval baseline depends on a fixed error corpus, so its scores should track how well the corpus covers the causes of the errors under test. We test this directly on ImageNet-R / CLIP by varying the share of style sentences in the corpus, namely the sentences matching ImageNet-R's controlled cause. Starting from the original corpus, where style sentences make up 14.0%, we either add irrelevant sentences as a size control, lowering the share to 9.3%, or inject style error explanations written with reference to the misclassified images under test, raising it to 42.7% and 57.0%. We also report how often the retriever picks a style sentence.

Table 9 shows that retrieval tracks coverage on every axis (LLM Match 0.24→0.47, CEI 11→33, style pick 40%→85%), while SciTX is unaffected, since the corpus holds pre-written explanations that only the baseline retrieves. The 57.0% condition is deliberately favorable to retrieval; the injected explanations were written while consulting the very errors under test, and a corpus deployed before those errors occur cannot contain such sentences. Retrieval's high CEI there (32.71) is therefore expected. Even in this condition, its LLM Match saturates near 0.5, since generic style sentences describe the factor only partially and earn the rubric's 0.5 credit, and stays below SciTX. The 9.3% condition shows the opposite direction; as relevant sentences become harder to retrieve, every axis falls below the original corpus. Retrieval therefore performs well only when the corpus already contains the causes of the errors and the retriever reliably picks them, whereas SciTX is robust to corpus coverage.

## C.3 Isolating the Editor's Contribution to CEI

Image editors redraw more than the instructed change, so an edit could raise the ground-truth probability through a generic quality gain rather than by neutralizing the described cause. We isolate the editor's own contribution on ImageNet-R / ViT with the FLUX editor, using two controls. **Enhance** replaces the explanation-derived instruction with "enhance the image quality," measuring the generic redraw effect alone. **Shuffled** runs the full pipeline with an explanation taken from a different image, namely a real, image-specific edit aimed at a mismatched cause.

Table 10: **Editing-confound ablation (ImageNet-R / ViT, FLUX editor).** A quality-enhancement edit alone barely shifts the prediction, and an explanation from a different image recovers only part of the matched effect.

| Intervention | CEI |
| --- | --- |
| Enhance | 1.07 |
| Shuffled | 6.94 |
| SciTX | **13.63** |

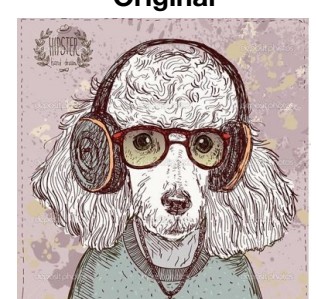
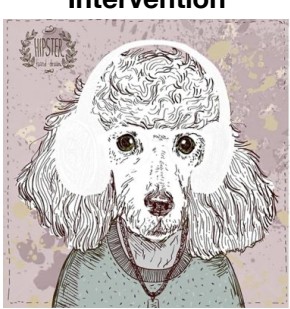

(a) Whimsical, snail-like tail → "Replace the ostrich's tail with a natural feather pattern."

(b) Hipster aesthetic → "Remove the headphones and glasses from the toy poodle."

Figure 10: **Abstract hypotheses compiled into concrete edit instructions (ViT / FLUX).** Each case pairs an abstract explanation with the concrete edit instruction it compiles into.

Table 10 shows that quality enhancement alone barely moves the prediction (1.07 against SciTX's matched 13.63). Shuffled reaches 6.94; this is a conservative control on ImageNet-R, whose errors share the style factor, so a shuffled explanation is often still roughly on-cause, yet matched explanations nearly double it. CEI therefore tracks whether the explanation names the correct cause, not the editor's redraw.

### C.4 Compiling Abstract Hypotheses into Edit Instructions

SciTX has no separate module or prompt for abstract or affective hypotheses. The edit-instruction compilation prompt (Appendix A) requires every edit to be a content-based visual change and forbids generic edits, so an abstract cue is compiled into a concrete modification of the object or scene that carries it. Fig. 10 shows two such cases from the ViT / FLUX logs. An explanation attributing the error to the whimsical, snail-like design of an ostrich's tail compiles into "Replace the ostrich's tail with a natural feather pattern," and one attributing a hipster aesthetic to a toy poodle compiles into "Remove the headphones and glasses from the toy poodle." In the CLIP and SigLIP settings the intervention is a prompt rewrite, so abstract concepts are expressed directly in text.

### C.5 Reliability of LLM Match

LLM Match is a rubric-based judgment rather than an open-ended quality assessment. Given the dataset-specific factor criterion, each judge assigns 1.0, 0.5, or 0.0 by whether the explanation describes the dataset's controlled factor, which makes the task closer to semi-mechanical factor matching than to subjective quality rating. Its reproducibility reflects this. The three judges, drawn from independent model families (Sec. 4.1.1), yield a majority verdict on 95.6% of samples, and removing any single judge leaves the method ranking essentially unchanged (Kendall $\tau \geq 0.963$, top method preserved in 26 of 27 setting × removed-judge checks).

To calibrate LLM Match against human annotations, we re-analyze the span highlights from our user study (Fig. 8), where participants marked good and bad parts of each explanation. We treat a good highlight on

the span naming the dataset's factor as a human judgment that the explanation identifies the intended factor, the same construct LLM Match measures. On SciTX explanations, LLM Match agrees with this human judgment in 77% of cases (23/30; Spearman $\rho = 0.47$), with active conflicts in only 3 of 30. The conflicts trace to an input asymmetry. Participants judged with the image in view, whereas the LLM judges see only the explanation text, so an explanation that names the right factor but grounds it incorrectly in the image is credited by the judges and rejected by humans. This image-grounding axis is exactly what CEI verifies, which is why we report the two metrics jointly.

### C.6 Disagreement between CEI and Human Judgments

We compare CEI with the human span highlights over the 120 explanations from our user study (Sec. 4.3) and examine the disagreement cases individually. Disagreements occur in 19 of 120 cases (16%). The dominant pattern, explanations humans endorse but CEI scores low (14 cases), concentrates on causes that image editing cannot reverse. These include 3D viewpoint changes (e.g., the compiled edit *"tilt the knife to show its side profile,"* which a 2D editor cannot execute), causes spanning the whole scene where a local edit leaves the context intact, and the narrow dynamic range of the prompt intervention. The reverse direction occurs in only 5 of 120 cases, where the MLLM generates an effective intervention even though the explanation itself is generic, so the prediction shifts while participants penalize the text. Both directions are properties of the intervention protocol, not of explanation quality, and are why we pair CEI with LLM Match rather than reporting either alone.

### C.7 Qualitative Results

More samples of generated explanations from baselines and ours are given. See Fig. 11 for ImageNet-D, Fig. 12 for ImageNet-R, and Fig. 13 for ObjectNet.

## D Human Study

Recognizing that mathematically optimized metrics do not always translate into good explanations for humans (Keane et al., 2021; Warren et al., 2023), we conducted a human study to evaluate the practical helpfulness of generated error explanations. We use 30 randomly selected misclassified samples, 10 each from ImageNet-R, ObjectNet, and ImageNet-D. To ensure a fair comparison, we selected samples with a high harmonic mean of CEI across all methods. Each participant evaluated a total of 15 images, with 5 samples drawn from each dataset.

As illustrated in Fig. 14, the evaluation interface provides participants with instructions, the input image, the ground truth, the model's prediction, and four candidate explanations. To mitigate bias, both the order of samples and the explanations were randomized. Participants were asked to rank the explanations across five dimensions without ties:

- Overall Helpfulness: Assessing which explanation is most helpful for understanding the error.

- Factuality: Evaluating the factual accuracy and reliability of the explanation.

- Information Density: Identifying which explanation is more concise yet information-dense.

- Specificity: Determining which explanation provides the most specific and relevant details about the model's error.

- Actionability: Judging which explanation provides the most actionable and useful insights for model improvement.

We recruited 30 AI practitioners via Prolific, a platform specialized in high-quality human data collection. Participants were compensated at a rate of £10/h, and the average completion time for the study was 57 minutes. Figure 15 shows the result; the ratio of rank-1 of ours is higher than others.

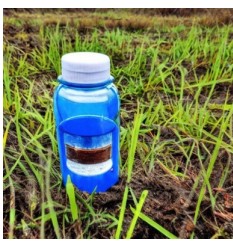

**Ground Truth:** Pill bottle
**Prediction:** Water bottle

**Retrieval**
The ground truth label for the image is actually incorrect.

**CamMLLM**
The model focused on the transparent part of the bottle, which is more characteristic of a water bottle, while ignoring the pill bottle's shape and contents, which are crucial for identifying it.

**ChangeMLLM**
The model might have been misled by the bottle's shape and the presence of a cap, which are common features of both pill bottles and water bottles. However, the content inside the bottle in the first image is distinct from typical water bottle contents, which could have confused the model.

**Ours**
The model misclassified the image because of the visual similarity between a pill bottle and a water bottle, particularly the blue color and the white cap. The background, which is a grassy field, might have led the model to associate the image more with a water bottle due to its common outdoor usage.

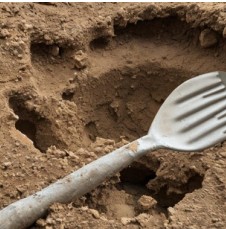

**Ground Truth:** Spatula
**Prediction:** Shovel

**Retrieval**
Only the skeletal remains of the object are shown.

**CamMLLM**
The model focused on the handle of the shovel, which is a common feature for both shovels and spatulas. However, the GradCAM also highlighted the blade area, which is more characteristic of a spatula. The model's prediction was misled by the handle, as it is a shared feature between shovels and spatulas.

**ChangeMLLM**
The model likely confused the spatula's ridged edge with the texture of a shovel blade, leading to the incorrect classification.

**Ours**
The model's misclassification can be attributed to the context and background. The spatula is placed in a dirt pit, which might have led the model to associate it more with a tool used for digging, thus predicting 'shovel'. The visual similarity in texture and shape between the spatula and the shovel could also contribute to this confusion.

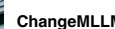

**Ground Truth:** Binder
**Prediction:** Envelope

**Retrieval**
Image is a print on a curved surface like a mug or shirt.

**CamMLLM**
The model focused on the yellow gradient pattern, which is not relevant to the true label 'binder'. The true label 'binder' would require attention to the functional shape and structure of the object, which was not highlighted by the GradCAM.

**ChangeMLLM**
The model may have been misled by the folded nature of the items, which can sometimes resemble envelopes due to their flat presentation. Additionally, the context of the first image being labeled as a 'binder' might have influenced the model's prediction.

**Ours**
The model likely misclassified the image as an 'envelope' due to the presence of the letter 'S' logo, which resembles the design often found on envelopes. The neat folding and the visible stitching along the edges also contribute to the envelope-like appearance, leading the model to confuse it with an envelope rather than a binder.

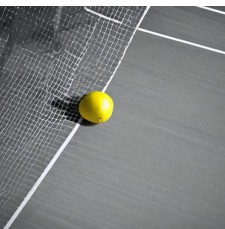

**Ground Truth:** Lemon
**Prediction:** Ping pong ball

**Retrieval**
Object is viewed from directly above, obscuring the side profile.

**CamMLLM**
The model focused on the bright yellow region near the center, which is consistent with the color of a ping-pong ball. However, this region is also characteristic of lemons, leading to the misclassification.

**ChangeMLLM**
The model may have been misled by the bright yellow color, which is common for both lemons and ping-pong balls, leading it to incorrectly classify the lemon as a ping-pong ball.

**Ours**
The model likely misclassified the lemon as a ping-pong ball due to the bright yellow color and round shape, which are common characteristics of both objects. The context of the image, showing a tennis court with a net, may have led the model to associate the yellow object with a ping-pong ball rather than a lemon.

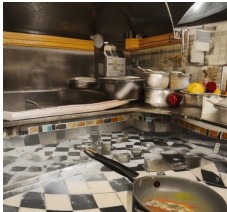

**Ground Truth:** Frying pan
**Prediction:** Stove

**Retrieval**
Typically co-occurring objects are missing from the scene.

**CamMLLM**
The model focused on the colorful checkered floor, which is irrelevant to the true label 'frying_pan'. The frying pan itself was not highlighted by the GradCAM, leading to the misclassification.

**ChangeMLLM**
The model might have been misled by the presence of the frying pan in the first image, which is a common kitchen tool often associated with stoves. However, the pan's position and the overall context of the image suggest that it is being used for cooking rather than the stove itself.

**Ours**
The model may have misclassified the frying pan as a stove due to the visual similarity in the checkered countertop and the industrial setting. The checkered pattern on the countertop could be mistaken for the tiled backsplash typically found behind a stove, leading the model to predict the presence of a stove.

Figure 11: Generated explanation from ImageNet-D samples.

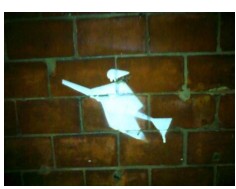

**Ground Truth:** Broom
**Prediction:** Military aircraft

**Retrieval**
Object is viewed from below, causing extreme foreshortening.

**CamMLLM**
The model focused on the geometric shapes and lines within the image, which are characteristic of a military aircraft, rather than the overall context or the presence of a broom. This led to the misclassification as the model did not recognize the broom's shape and structure.

**ChangeMLLM**
The model may have been misled by the white silhouette in the first image, which resembles the shape of a military aircraft, leading it to incorrectly predict 'military_aircraft' instead of 'broom'. The simplicity and distinctiveness of the broom in the second image likely helped the model recognize it correctly.

**Ours**
The model misclassified the image due to the strong visual similarity between the glowing silhouette of a military aircraft and the broom. The aircraft's streamlined, angular shape and the way it is positioned against the wall closely resemble the broom's form, leading the model to confuse them.

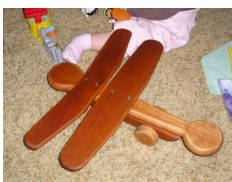

**Ground Truth:** Bell pepper
**Prediction:** Banana

**Retrieval**
An unnatural color filter alters the object's appearance.

**CamMLLM**
The model focused on the bright, glowing areas around the banana, which may have been interpreted as a sign of ripeness or sweetness, leading it to predict 'banana' instead of recognizing the distinct shape and color of the bell pepper.

**ChangeMLLM**
The model may have been trained on a dataset where the bell pepper was often depicted with a different color or shape, leading to confusion when presented with the yellow bell pepper costume.

**Ours**
The model may have confused the large, curved shape of the bell pepper with the similar contour of a banana. The vibrant yellow color and elongated form strongly resemble a banana, even though the texture and details are distinctly different.

**Ground Truth:** Dragonfly
**Prediction:** Canoe

**Retrieval**
Object is a plush toy lacking rigid structure.

**CamMLLM**
The model focused on the red-highlighted regions, which appear to be the body parts of the dragonfly, but it misclassified them as a canoe due to the similarity in shape and color gradient. The model did not consider the distinct features of the dragonfly's wings and body, which are crucial for identifying the true label.

**ChangeMLLM**
The model might have been trained on a dataset that did not include images of dragonflies, leading to incorrect classification.

**Ours**
The model's misclassification from 'dragonfly' to 'canoe' is likely due to the visual similarity between the wooden toy airplane and the canoe. Both objects have a streamlined, elongated shape with a single central body and two side structures resembling wings or oars. The simplicity and symmetry of the shapes, along with the lack of distinct features that differentiate a dragonfly from a canoe, contributed to the model's confusion.

**Ground Truth:** Goose
**Prediction:** Bathtub

**Retrieval**
Object pattern blends perfectly into the background (camouflage).

**CamMLLM**
The model focused on the colorful, abstract patterns on the wall, which are not relevant to the true label 'goose'. The model did not consider the presence of the goose, which was likely obscured by the vibrant background.

**ChangeMLLM**
The model may have been misled by the context of the image, assuming that the presence of a door and a tiled floor indicated a bathtub rather than recognizing the mural as the primary subject.

**Ours**
The model misclassified the goose as a bathtub due to the presence of bathroom-related elements such as the tiled floor, shower, and sink in the background. The wall art depicting geese did not provide enough distinguishing features to override the contextual clues suggesting a bathroom setting.

**Ground Truth:** Goose
**Prediction:** Bathtub

**Retrieval**
A different species has evolved to mimic the appearance of the object.

**CamMLLM**
The model focused on the body and wings of the insect, which are common features in both electric guitars and cockroaches. However, the presence of antennae and the overall shape of the insect are distinctive features of a cockroach that the model did not emphasize, leading to the misclassification.

**ChangeMLLM**
The model may have been misled by the variety of colors and patterns in the first image, which could have distracted it from recognizing the commonality of the cockroach shapes. It might have focused more on the unique visual elements rather than the overall form.

**Ours**
The model's misclassification can be attributed to the presence of artistic and decorative elements in the cockroach designs that resemble the stylized and colorful features of the electric guitars. The vibrant colors, intricate patterns, and overall aesthetic of both the cockroaches and guitars share similar artistic qualities, leading the model to confuse them.

Figure 12: Generated explanation from ImageNet-R samples.

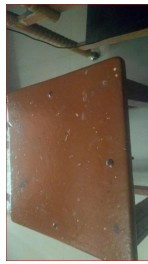

**Ground Truth:** Bench
**Prediction:** Binder

**Retrieval**
Camera is too close, showing only texture without shape.

**CamMLLM**
The model may have been misled by the texture and color patterns in the central and upper parts of the image, which are not indicative of a bench, leading to the incorrect prediction of 'Binder (closed).'

**ChangeMLLM**
The model likely failed to recognize the overall shape and context of the object in the first image due to its close-up nature and lack of surrounding elements that typically define a bench.

**Ours**
The model misclassified the image due to the similarity in the flat, rectangular shape and the presence of a metal fastener, which are common features of binders. The reddish-brown color and the texture of the surface also contributed to the confusion, as they resemble the appearance of a closed binder.

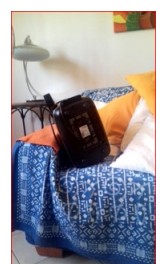

**Ground Truth:** Toaster
**Prediction:** Portable heater

**Retrieval**
Camera is too close, showing only texture without shape.

**CamMLLM**
The model focused on the control panel, which is characteristic of a portable heater, while ignoring the key feature of the toaster, i.e., the slots for bread.

**ChangeMLLM**
The model might have been misled by the shape and size of the device, which resembles a toaster, but the presence of a label and the context of the room (a bed) could have confused it into predicting it as a portable heater.

**Ours**
The model misclassified the object due to the similar rectangular shape of the appliance and the lack of distinguishing features that differentiate a toaster from a portable heater. The bed setting might have contributed to the model's uncertainty, as toasters are typically kitchen appliances, not bedroom items.

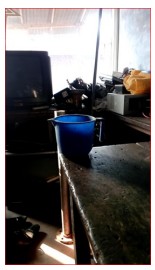

**Ground Truth:** Measuring cup
**Prediction:** Bucket

**Retrieval**
Image is overexposed, washing out the object's contours.

**CamMLLM**
The model focused too broadly on the bucket's body and handle, missing the distinctive spout characteristic of a measuring cup, leading to the misclassification.

**ChangeMLLM**
The model might have been misled by the handle on the blue container in the first image, which resembles a bucket's handle, leading it to incorrectly classify the object as a bucket instead of recognizing the measurement markings typically associated with a measuring cup.

**Ours**
The model's misclassification can be attributed to the lack of distinct features that differentiate a measuring cup from a bucket. The blue object in both images appears to have a handle and a spout-like structure, which are common to both utensils. However, the context of the first image, featuring a cluttered workspace with various electronic equipment, suggests a more utilitarian environment where buckets could be commonly used for storage or water collection. In contrast, the second image shows a single blue object without additional context, leading the model to categorize it as a bucket due to its shape and lack of specific measuring markings.

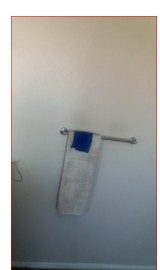

**Ground Truth:** Bath towel
**Prediction:** Soap dispenser

**Retrieval**
Object is viewed from below, causing extreme foreshortening.

**CamMLLM**
The model misclassified the image due to its strong focus on the soap dispenser, overlooking the presence of the bath towel.

**ChangeMLLM**
The model may have been misled by the towel's appearance alone without recognizing the context of its placement, which is typically found in a bathroom setting.

**Ours**
The model likely misclassified the object due to the presence of a blue fabric piece at the top of the towel, which resembles the spout of a soap dispenser. The texture and shape of this piece, combined with the overall appearance of the object, may have led the model to confuse it with a soap dispenser.

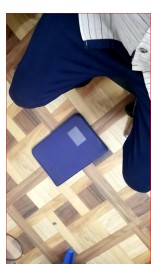

**Ground Truth:** Binder
**Prediction:** Weight

**Retrieval**
Incorrect white balance makes the object look unnatural.

**CamMLLM**
The model focused on the blue fabric-like areas due to their color similarity with the exercise equipment, leading it to predict 'Weight (exercise)' instead of the correct 'Binder (closed).'

**ChangeMLLM**
The model likely misclassified the first image because it included a person's legs, which can be mistakenly associated with a weight exercise context, despite the actual object being a closed binder.

**Ours**
The model misclassified the closed purple binder as a 'Weight (exercise)' due to the presence of a blue folder-like object on the wooden floor, which resembles a weight plate often used in gym settings. The blue folder's shape and color might have been mistaken for a weight plate, leading to the incorrect prediction.

Figure 13: Generated explanation from objectnet samples.

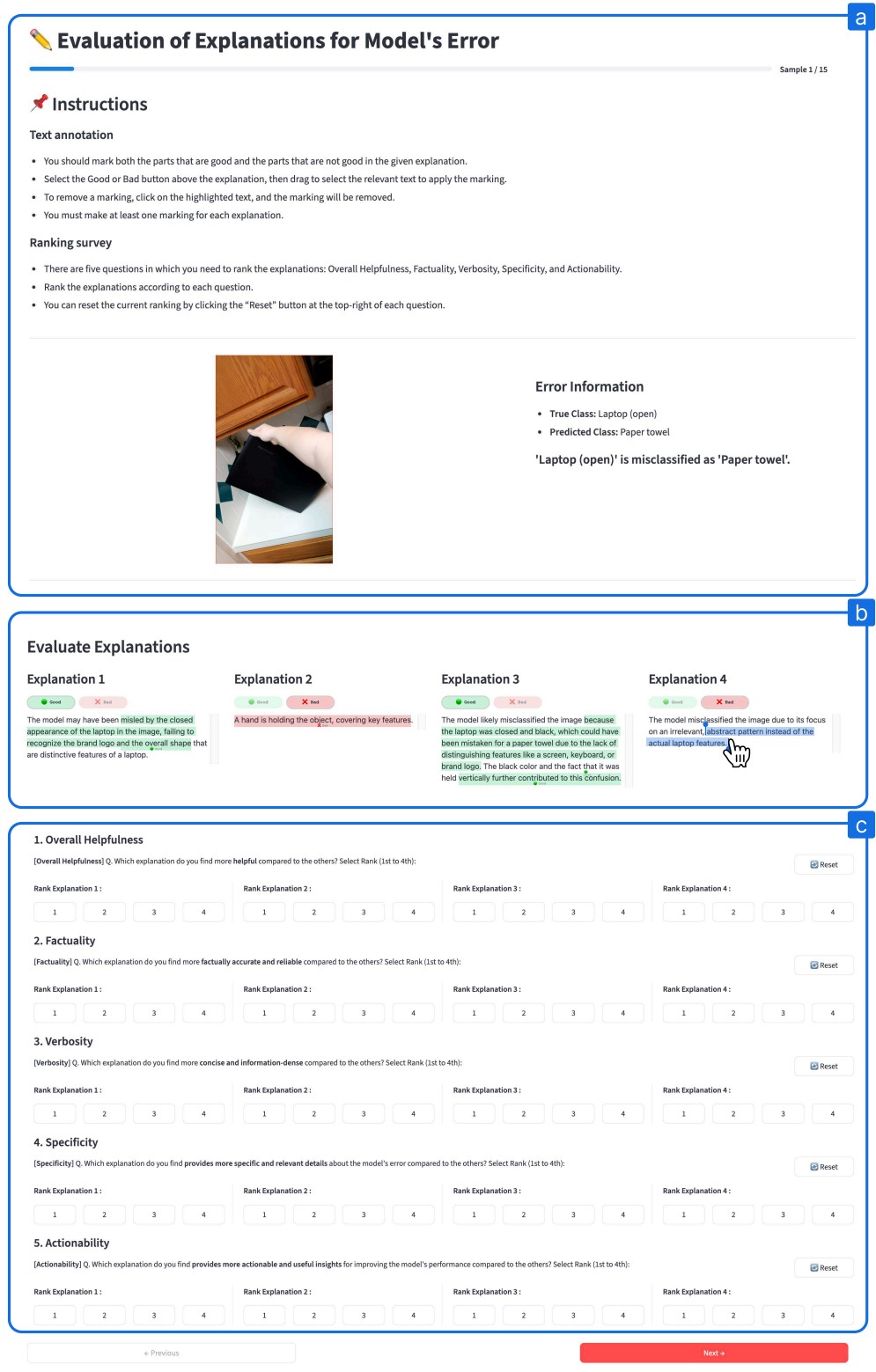

Figure 14: Interface for human evaluation. (a) The task instruction and an example showing the ground-truth and predicted classes. (b) An interface that allows participants to directly annotate helpful and unhelpful parts of the explanation. (c) An interface for ranking explanations across evaluation criteria.

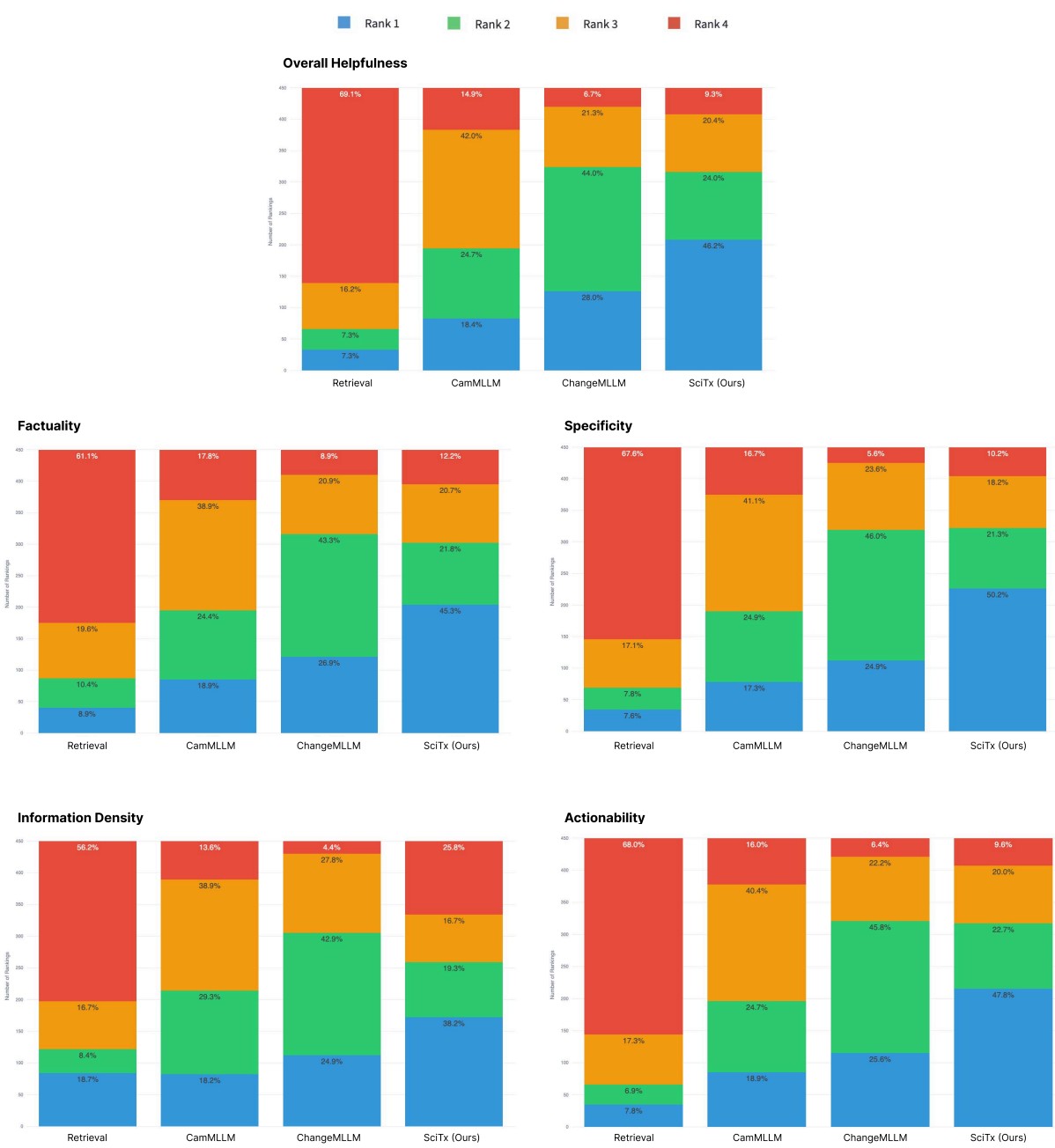

Figure 15: Detailed human evaluation results for five categories

