# OpenReview forum: "Hypothesize and Verify: Natural-Language Explanations of Vision Model Errors"
_TMLR — Under review for TMLR_

### Review · Reviewer_xtCp · 2026-06-21

**Summary Of Contributions:**

This paper proposes a way to generate natural language explanations (NLEs) of failures of image classifiers (such as ViTs).  The method, in a nutshell, fetches semantically similar examples and observes the classifier result, converting effects into counterfactual interventions. It returns the explanation with the larges improvement.
It also proposes a benchmark of misclassified images with an LLM-as-a-judge protocol to evaluate.

## Strengths
* This paper nicely combines retrieval, hypothesis generation, and counterfactual validation.
* It contains enough empirical evaluation including multiple classifiers, datasets, ablations, robustness studies, and human evaluation.

## Weaknesses
* Combines existing components (retrieval, MLLM generation, image editing, counterfactual testing) rather than producing a fundamentally new method / artefact.
* The computational cost of generating multiple hypotheses and validating each through counterfactual editing is not discussed.
* CEI is also used during inference for explanation selection, making it difficult to separate improvements in explanation quality from optimization toward the evaluation metric. Furthermore, the relationship between CEI and explanation faithfulness is assumed rather than thoroughly validated.

**Additional Comments:**

none

**Audience:**

Yes

**Audience Explanation:**

The paper addresses an increasingly relevant problem: generating faithful natural-language explanations of model failures for non-expert users. As LLM-based debugging interfaces become more common, understanding whether generated explanations correspond to actual model behavior is an important research direction.

**Broader Impact Concerns:**

Broader Impact Statement existent

**Claims And Evidence:**

Yes

**Claims Explanation:**

Yes, mostly.

The empirical evaluation is extensive and generally supports the main experimental claims. The paper evaluates across multiple datasets, classifiers, MLLM backbones, image editors, and includes both automated metrics and human evaluation. The authors also perform several useful ablations.

However, several claims would benefit from stronger evidence:

1. The central assumption that higher CEI corresponds to more faithful explanations is plausible but not rigorously validated. An intervention increasing the correct-class probability does not necessarily imply that the explanation captures the actual causal mechanism used by the classifier. CEI demonstrates that the proposed intervention is effective, not necessarily that the explanation identifies the *mechanistic* process or evidence the classifier originally relied upon. Counterfactual edits may simultaneously modify multiple visual attributes or move the image toward an easier region of the data distribution, increasing classifier confidence without validating the proposed causal explanation. A more detailed discussion of this distinction would strengthen the paper.

2. LLM Match largely measures whether explanations mention the benchmark's controlled corruption factor. This is appropriate for NEMO but may overestimate explanation quality on naturally occurring failures where no single known factor exists. Giving that the benchmark focuses on datasets with known synthetic perturbations, leaves open whether SciTX generalizes to more realistic model failures.

3. Since CEI is both the selection criterion and one of the reported metrics, the paper appropriately reports matched-selection baselines.

3. Figure 8 provides qualitative insight into which explanation fragments humans find convincing or unconvincing. However, these annotations should not be interpreted as evidence of faithfulness, since annotators do not have access to the classifier's internal decision process and therefore cannot determine whether the highlighted evidence actually drove the prediction. There is plenty of literature about plausibility versus faithfulness, for example see this survey: https://arxiv.org/abs/2402.04614

Overall, I find the experimental evidence convincing for demonstrating improvements on the proposed benchmark, but somewhat less convincing regarding the broader claims about faithful explanation generation.

**Requested Changes:**

1. Better justify CEI as a faithfulness measure. The paper currently assumes that if an intervention derived from an explanation improves classifier confidence, then the explanation is faithful. However, interventions may alter multiple image properties simultaneously or simply produce an easier example, without demonstrating that the explanation corresponds to the evidence originally used by the classifier. The assumptions and limitations into the mechanistic insights of CEI should be discussed more carefully.
2. The paper would benefit from a calibration of the proposed LLM Match metric against human annotations. While averaging three independent LLM judges reduces model-specific bias, it remains unclear how well LLM Match agrees with human judgments on whether an explanation identifies the intended corruption factor.
2. Clarify the algorithmic novelty. The paper combines retrieval, MLLM generation, counterfactual image editing, and explanation selection into an effective pipeline. It would help readers if the paper more clearly articulated which component constitutes the primary methodological contribution beyond this integration.
3. Report inference cost (runtime, number of generated hypotheses, image edits, and MLLM calls) compared with the baselines.
4. Include additional discussion of situations where CEI may fail, for example when imperfect image editing introduces unintended changes.
5. Analyze disagreement cases between CEI and human judgments.
6. Discuss how sensitive the framework is to the choice and quality of the image editing model.

---

> ### Author Response · Authors · 2026-07-15
> **Response to Reviewer xtCp**
>
> We appreciate the reviewer’s positive assessment and constructive feedback. We address each point below, with all revisions in the manuscript highlighted in blue.
>
> **Q1. Justify CEI as a faithfulness measure, and discuss its assumptions and limitations.**
>
> We acknowledge that the scope of CEI requires clearer articulation. CEI employs a counterfactual test, where a positive score is evidence that the factor identified in the explanation is causally relevant to the error.
>
> - **Assumptions and limitations (Section 3.3).** The newly added “What CEI establishes” paragraph clarifies that a high CEI score provides evidence of causal relevance, not a mechanistic explanation. It also identifies the two failure modes noted by the reviewer: edits that modify multiple attributes simultaneously and edits that simply produce an easier image. Appendix C.3 quantifies these issues, showing that a quality-enhancement edit alone results in a low CEI (1.07).
> - **Revision of overstated claims.** Statements such as “Grounded in the model’s actual failure mechanism” (Introduction) and “thereby evaluating its faithfulness” (Related Work) have been removed or revised for accuracy.
>
>
> **Q2. Calibrate LLM Match against human annotations.**
>
> LLM Match operates as a rubric-based evaluation rather than an open-ended quality assessment.
>
> - **High agreement among judges**. Three judges from independent families yield a majority verdict on 95.6% of samples. Excluding any single judge does not substantially alter the method ranking (Kendall τ ≥ 0.963; the top method is preserved in 26 of 27 checks).
> - **Agreement between LLM Match and human annotations (77%).** We re-analyzed span highlights from our user study, interpreting a good highlight on the span naming the dataset’s factor as a human judgment of the same construct measured by LLM Match. LLM Match aligns with human judgments on SciTX explanations in 77% of cases (23/30), with active conflicts in 10% (3/30).
> - **Input asymmetry.** Participants evaluated explanations with the image visible, while LLM judges considered only the explanation text. Consequently, explanations that identify the correct factor but misattribute it within the image are credited by LLM judges but rejected by human evaluators. This distinction in image-grounding is precisely what CEI assesses, motivating our joint reporting of both metrics.
>
> We have added both analyses to Appendix C.5, “Reliability of LLM Match,” and referenced them in Section 3.4.
>
> **Q3. Clarify the primary methodological contribution beyond the integration.**
>
> The primary methodological contribution is the hypothesize-and-verify procedure, which transforms each free-form natural-language hypothesis into an executable counterfactual intervention and tests it against the target model. Observation retrieval and MLLM generation serve as supporting components constructed from standard tools; their individual contributions are detailed in Table 2 (CEI selection module ablation) and Table 5 (observation-retrieval ablation). We have revised the contribution statement in the Introduction to reflect this.

---

> > ### Author Response · Authors · 2026-07-15
> >
> > **Q4. Report inference cost (runtime, number of generated hypotheses, image edits, and MLLM calls) compared with the baselines.**
> >
> > We measured the per-sample wall-clock cost for each explainer in the ViT / FLUX image-editing setting, which represents the most computationally intensive configuration:
> >
> >
> > | Explainer                                                        | Time / sample | Main cost                                            |
> > | ---------------------------------------------------------------- | ------------- | ---------------------------------------------------- |
> > | Retrieval                                                        | 0.02s         | embedding similarity search only (no VLM, no editor) |
> > | CamMLLM                                                          | 1.82s         | 1 VLM call (1.74s)                                   |
> > | ChangeMLLM                                                       | 3.35s         | 1 VLM call (3.33s)                                   |
> > | **SciTX**                                                        | **158.5s**    | broken down below                                    |
> >
> > **Table. Per-sample wall-clock cost (single NVIDIA RTX A6000).**
> >
> >
> >
> > | SciTX stage                    | Calls × time | Time / sample | Share |
> > | ------------------------------ | ------------ | ------------- | ----- |
> > | Contrastive caption generation | 10 × 1.09s   | 10.9s         | 6.9%  |
> > | Hypothesis generation (M=5)    | 5 × 2.37s    | 11.9s         | 7.5%  |
> > | Edit-instruction               | 5 × 1.74s    | 8.7s          | 5.5%  |
> > | FLUX editing (verification)    | 5 × 25.4s    | 127.0s        | 80.1% |
> >
> > **Table. SciTX per-sample breakdown by stage, with the number of calls at each stage.**
> >
> >
> > - **SciTX incurs the highest computational cost (158.5 seconds per sample).** The M=5 FLUX edits constitute 80% of this total, resulting in a cost that scales approximately linearly with M. These edits can be executed in parallel across GPUs. In the CLIP and SigLIP settings, the intervention involves prompt rewriting, eliminating the diffusion cost.
> > - **M trades cost for accuracy.** At M=1, the per-sample cost decreases to 13.3 seconds (a twelvefold reduction), while LLM Match performance declines by only 0.071 and remains superior to all baselines.
> > - **The cost is a one-time analysis expense.** SciTX is an offline diagnostic tool run once over a fixed error set, not a serving-time component.
> >
> > The stage table provides the requested counts: hypotheses (M=5), image edits (5), and MLLM calls (10 captions, 5 hypotheses, 5 edit instructions). This analysis has been incorporated into the main text as a new “Computational Cost” subsection (Section 4.2.8).
> >
> > **Q5 & Q6. Discuss situations where CEI may fail, and analyze disagreement cases between CEI and human judgments.**
> >
> > To identify instances where CEI diverges from human judgments, we compared CEI scores with human span highlights across 120 explanations from our user study and analyzed disagreement cases individually. Disagreements occur in 19 of 120 cases (16%) and fall into two categories.
> >
> > - **CEI low, humans accept (14 of 19).** These cases primarily involve causes that image editing cannot address, such as 3D viewpoint changes (for example, the edit “tilt the knife to show its side profile,” which a 2D editor cannot perform), scene-wide causes where local edits do not alter the context, and the limited dynamic range of the prompt intervention.
> > - **CEI high, humans reject (5 of 19).** In these instances, the MLLM produces an effective intervention despite the explanation being generic, resulting in a prediction shift while participants penalize the explanation text.
> >
> > We have added this analysis as Appendix C.6 and pointed to it in Section 4.3.
> >
> > **Q7. Discuss how sensitive the framework is to the choice and quality of the image editing model.**
> >
> > Sensitivity varies depending on the aspect of the editor’s influence.
> >
> > - **The method ranking is insensitive to the editor choice.** All methods are evaluated using the same editor, and replacing FLUX.1-Kontext-dev with Qwen Image Edit preserves the ranking (Table 3). Absolute CEI values do differ, since the editor acts as the measuring instrument; we compare absolute values only within a fixed setup, and the revision now states this in Section 4.2.3.
> > - **Editor quality affects the absolute scale of CEI scores but not the comparative ranking.** Our results do not determine which editor is superior. In principle, a more capable editor reduces the editability bias discussed in Appendix C.6, as fewer correct explanations are discarded due to hard-to-execute causes. Appendix C.3 demonstrates that the editor’s direct contribution to CEI is low (1.07), so improvements in editor quality enhance the test’s precision rather than artificially increasing scores.

---

> > > ### Comment · Reviewer_xtCp · 2026-07-18
> > > **Reviewer Response**
> > >
> > > Thanks a lot, you really went through all suggestion points and took action on them. Well done, much appreciated!

---

### Review · Reviewer_w7ox · 2026-06-23

**Summary Of Contributions:**

### Summary

This paper addresses key challenges in current visual classification models, such as the lack of natural language interfaces for error diagnosis, the limitation of existing retrieval methods to fixed error corpora, and the susceptibility of MLLMs to hallucination when generating direct explanations. The main contributions of this work are twofold: First, it establishes the NEMO benchmark and evaluation protocol based on 1,200 misclassified images spanning three robustness benchmarks, utilizing an LLM-as-a-judge mechanism to evaluate whether the generated explanations identify the confounding factors leading to the misclassification. Second, it proposes SciTX, a method that establishes contrastive observations via reference sample retrieval, generates candidate causal hypotheses, and executes counterfactual interventions using image editing or prompt rewriting. Hypotheses that yield the maximum correction amplitude on the model's prediction probability are then selected as the final explanation based on a proposed CEI metric. Extensive experiments and user studies demonstrate that SciTX outperforms existing baselines across multiple automated and human evaluation metrics.

### Strengths

- The paper introduces the NEMO evaluation protocol, which is designed for evaluating free-text natural language explanations of vision model errors, accompanied by an automated LLM Match scoring pipeline.
- The paper proposes the SciTX method. It incorporates counterfactual interventions to explicitly verify the faithfulness of explanations. This ensures that the generated language explanations are truly anchored in the target classifier's actual failure mechanisms, rather than relying solely on the parametric knowledge of the MLLM.
- The paper introduces the CEI metric and validates the effectiveness of SciTX through cross-dataset evaluations on the NEMO benchmark.

### Weaknesses & Questions

1. The core of this paper centers on utilizing counterfactual editing to validate natural language hypotheses regarding model errors, a concept that has seen related exploration in the field, such as in [1–4]. Where does the core technical advantage of SciTX truly lie, e.g., in the prompt structure used by the MLLM to generate hypotheses, or in the CEI-driven verification module? If the internal causal selection module of SciTX were integrated into conventional baselines, what would constitute the unique merits of SciTX itself?

[1] Zhang, X., Liu, Z., Zhang, Y., Hu, X., & Shao, W. (2026). Retroagent: From solving to evolving via retrospective dual intrinsic feedback.

[2] Naiseh, M., Simkute, A., Zieni, B., Jiang, N., & Ali, R. (2024). C-XAI: A conceptual framework for designing XAI tools that support trust calibration.

[3] Kim, S., Oh, J., Lee, S., Yu, S., Do, J., & Taghavi, T. (2023). Grounding counterfactual explanation of image classifiers to textual concept space.

[4] Wang, X., Wang, Z., Weng, H., Guo, H., Zhang, Z., Jin, L., ... & Ren, K. (2023). Counterfactual-based saliency map: Towards visual contrastive explanations for neural networks.

2. The proposed method relies heavily on generative image editing models to execute the counterfactual intervention $\mathcal{G}(Img, \tau)$. However, upon receiving editing instructions, current generative editing tools often modify the background while simultaneously redrawing or smoothing the edges, textures, and lighting of the foreground subject. This global feature redrawing might indirectly enhance the overall clarity of the image or magnify prominent class-specific features, thereby artificially boosting the target model's prediction probability for the true label $P(y|s_e)$. Consequently, the probability gains measured by the CEI metric may not stem from explanation $e$ accurately identifying the causal failure factor, but rather from a confounding quality enhancement or feature amplification effect introduced by the editing model itself. The paper lacks quantitative ablations or fidelity constraints targeting this severe confounding variable.

3. In practice, how does the MLLM translate highly abstract emotional semantics into a concrete, executable instruction prompt $\tau$ limited to 10 words? Furthermore, how does the image editing model intervene on an abstract concept in the pixel space without altering the object's contours or class identity? The paper only provides prompts for physical feature modifications, leaving a lack of reproducible implementation details regarding the intervention generation mechanisms.

4. For every error sample, SciTX requires one retrieval step, $K$ caption generation steps, and notably, an inner-loop cycle across $M$ candidate hypotheses. Within this loop, evaluating each hypothesis $h_j$ requires invoking a large instruction-following image diffusion model for forward inference to compute $\text{CEI}(h_j)$. Frequent calls to such diffusion models introduce significant computational latency. Please provide a detailed evaluation of the computational overhead and runtime costs.

5. If the system is constrained to $M=1$ (i.e., skipping the causal verification and directly outputting the first hypothesis generated by the MLLM as the final explanation), by how much does the LLM Match performance degrade? This ablation is crucial for demonstrating whether the superiority of SciTX stems from the contrastive retrieval design during the hypothesis stage, or from the subsequent experimental verification.

**Audience:**

Yes

**Audience Explanation:**

This work lies at the intersection of XAI and MLLM benchmarking, both of which are highly active fields of interest for the TMLR community.

**Broader Impact Concerns:**

The submission includes a dedicated "Broader Impact Statement" section that properly states its natural-language explanations serve as a diagnostic starting point rather than verified facts, integrating safely within human-in-the-loop workflows. The section may overlook two critical sociotechnical implications: first, it fails to address how the authoritative fluency of SciTX's textual explanations might induce automation bias or misplaced trust calibration among non-expert users, leading to flawed downstream debugging decisions; second, the implicit dual-use risk is insufficiently examined, as the pipeline's core mechanism provides an optimization blueprint that could be inverted by adversaries to systematically discover minimal, deceptive modifications to bypass vision classifiers.

**Claims And Evidence:**

Yes

**Claims Explanation:**

The main claims regarding the baseline performance of the proposed NEMO benchmark and the conceptual framework of SciTX are supported by structured empirical evaluations, cross-model verification, and a comprehensive human study with 30 AI practitioners. However, several core algorithmic claims and methodological assumptions are only partially supported; more discussion about the points mentioned in Weaknesses is required, e.g., ablation of the verification core, evaluation of computational efficiency, etc.

**Requested Changes:**

Please see "Summary Of Contributions" for details.

---

> ### Author Response · Authors · 2026-07-15
> **Response to Reviewer w7ox**
>
> We appreciate the reviewer’s detailed and thoughtful questions. Our responses to each are provided below, with all changes in the revised manuscript marked in blue.
>
> **Q1. Clarify the core technical advantage over [1–4], and the unique merit beyond the CEI-driven selection module.**
>
> None of [1-4] validates an open-ended natural-language hypothesis regarding another model’s error using an input-space counterfactual intervention. The overlap with each prior work occurs along a different methodological axis.
>
> |                | Output                                                         | Role of counterfactual                                           | Role of language                                        |
> | -------------- | -------------------------------------------------------------- | ---------------------------------------------------------------- | ------------------------------------------------------- |
> | [1] RetroAgent | improved policy (RL)                                           | none (hindsight rewards)                                         | Intrinsic Language Feedback, lessons for its own policy |
> | [2] C-XAI      | interface design process (complementary layer)                 | not central                                                             | guidelines for presenting existing explanations         |
> | [3] CounTEX    | concept importance scores                                      | **produces** the explanation (latent perturbation)               | concept vocabulary                                      |
> | [4] CCE        | saliency heatmaps, the cause left to the user’s interpretation | **produces** the explanation (pixel perturbation)                | none                                                    |
> | **SciTX**      | free-form explanation of another model’s error                 | **verifies** the explanation (input-space edit / prompt rewrite) | open-ended hypothesis, delivered to the user            |
>
>
> - **Discussion of CounTEX [3], the closest work.** Both measure how a concept expressed in text influences the model’s prediction. However, SciTX differs in that its concept is an open-ended, sample-specific hypothesis rather than a selection from a predefined concept library. Additionally, SciTX intervenes not only through the prompt but also directly in the image for non-VLM classifiers.
> - **Unique merit beyond the CEI-driven selection module.** SciTX outperforms all baselines even in the absence of the selection module (Q5). Its remaining advantage lies in hypothesis generation grounded in observations from correctly classified samples of both the ground-truth and predicted classes, a capability that the selection module alone cannot replicate.
>
> We have added [3] and [4] to the Related Work with this distinction.
>
> **Q2. Ablate the editing confound: quality enhancement by the editor could inflate CEI.**
>
> | ImageNet-R / ViT                                                      | CEI      |
> | --------------------------------------------------------------------- | -------- |
> | Enhance                                                               | 1.07     |
> | Shuffled                                                              | 6.94     |
> | SciTX                                                                 | 13.63    |
>
> **Table. Editing-confound ablation (ImageNet-R / ViT, FLUX editor).**
>
>
> We conducted a quantitative ablation to isolate the editor’s contribution to CEI within the ViT / FLUX image-editing setting.
>
> - **Quality enhancement alone has minimal impact on CEI (1.07)**. In the enhanced control, the explanation-derived instruction is replaced with “enhance the image quality,” resulting in a CEI of 1.07, compared to SciTX’s matched value of 13.63.
> - **Using a different explanation recovers only half the effect (6.94).** The shuffled control applies the pipeline with an explanation sourced from a different image. This serves as a conservative control on ImageNet-R, where errors often share a style factor, making a shuffled explanation still approximately relevant. Nevertheless, matched interventions nearly double the effect.
>
> Collectively, these results demonstrate that CEI reflects whether the explanation identifies the correct cause, rather than the editor’s redraw. We have included this ablation and its discussion as Appendix C.3, and the image-editor robustness paragraph in Section 4.2.3 now references this addition.

---

> > ### Author Response · Authors · 2026-07-15
> >
> > **Q3. Explain how abstract or emotional semantics compile into concrete edit instructions.**
> >
> > There is no separate module or prompt for abstract or emotional hypotheses.
> >
> > - **Abstract cues are translated into concrete visual modifications.** The edit-instruction compilation prompt (Appendix A) requires each edit to be a content-based visual change. For example, in the ViT / FLUX logs, a whimsical, snail-like ostrich tail is rendered as “Replace the ostrich’s tail with a natural feather pattern,” while a hipster aesthetic on a toy poodle becomes “Remove the headphones and glasses from the toy poodle.”
> > - **Contours and class identity are preserved.** The instruction modifies the specific element embodying the abstract concept (such as the tail or headphones), rather than altering the entire object.
> >
> > We have added this discussion and the two examples as Appendix C.4 (Figure 10).
> >
> >
> >
> > **Q4. Report the computational overhead and runtime costs.**
> >
> > We measured per-sample wall-clock costs for each explainer in the ViT/FLUX image-editing setting, which represents the most computationally intensive configuration.
> >
> > | Explainer                                                        | Time / sample | Main cost                                            |
> > | ---------------------------------------------------------------- | ------------- | ---------------------------------------------------- |
> > | Retrieval                                                        | 0.02s         | embedding similarity search only (no VLM, no editor) |
> > | CamMLLM                                                          | 1.82s         | 1 VLM call (1.74s)                                   |
> > | ChangeMLLM                                                       | 3.35s         | 1 VLM call (3.33s)                                   |
> > | **SciTX**                                                        | **158.5s**    | broken down below                                    |
> >
> > **Table. Per-sample wall-clock cost (single NVIDIA RTX A6000).**
> >
> >
> > | SciTX stage                    | Calls × time | Time / sample | Share |
> > | ------------------------------ | ------------ | ------------- | ----- |
> > | Contrastive caption generation | 10 × 1.09s   | 10.9s         | 6.9%  |
> > | Hypothesis generation (M=5)    | 5 × 2.37s    | 11.9s         | 7.5%  |
> > | Edit-instruction               | 5 × 1.74s    | 8.7s          | 5.5%  |
> > | FLUX editing (verification)    | 5 × 25.4s    | 127.0s        | 80.1% |
> >
> > **Table. SciTX per-sample breakdown.**
> >
> >
> > - **SciTX incurs the highest computational cost among explainers (158.5 seconds per sample).** The M=5 FLUX edits account for 80% of this total, making the cost approximately linear in M, and the edits can be executed in parallel across GPUs. In the CLIP and SigLIP settings, the intervention consists of a prompt rewrite, eliminating the diffusion cost.
> > - **M trades cost for accuracy.** At M=1, the per-sample cost decreases to 13.3 seconds (a twelvefold reduction), while LLM Match declines by only 0.071 and still surpasses all baselines (Q5).
> > - **This computational cost represents a one-time analysis expense.** SciTX functions as an offline diagnostic tool, executed once over a fixed error set, rather than as a serving-time component.
> >
> > We have incorporated this analysis into the main text as a new “Computational Cost” subsection (Section 4.2.8), explicitly stating that runtime constitutes SciTX’s primary practical limitation.

---

> > > ### Author Response · Authors · 2026-07-15
> > >
> > > **Q5. Quantify the degradation without causal verification (M=1).**
> > >
> > > | Method (ImageNet-D / CLIP) | w/o Val. LLM Match | w/o Val. CEI | w/ Val. LLM Match | w/ Val. CEI   |
> > > | -------------------------- | ------------------ | ------------ | ----------------- | ------------- |
> > > | Retrieval                  | 0.147              | 3.31         | 0.150 (+0.003)    | 4.72 (+1.41)  |
> > > | CamMLLM                    | 0.522              | 5.55         | 0.513 (−0.009)    | 6.26 (+0.71)  |
> > > | ChangeMLLM                 | 0.373              | 11.92        | 0.532 (+0.159)    | 12.00 (+0.08) |
> > > | **SciTX**                  | **0.548**          | **12.01**    | **0.619**         | **19.27**     |
> > >
> > > **Table 2, completed. The SciTX “w/o Validation” cells (= M=1), previously “–”, are now filled.**
> > >
> > >
> > > - **The observed degradation is 0.071 (0.619 → 0.548).** At M=1, verification is omitted, and the first MLLM hypothesis is output directly.
> > > - **Even without verification, SciTX outperforms all baselines.** Its score of 0.548 surpasses the best baseline, even with validation (ChangeMLLM, 0.532). Thus, the primary advantage arises from the stages preceding verification, specifically contrastive observation retrieval and hypothesis generation.
> > > - **Verification still contributes.** It adds +0.071 LLM Match and most of the CEI margin (12.01 → 19.27) for SciTX, and also helps the baselines (ChangeMLLM +0.159).
> > >
> > > We have filled the previously empty SciTX “w/o Validation” cells in Table 2 (0.548 / 12.01) and added this interpretation to the fair-comparison paragraph (Section 4.2.2).

---

> > > > ### Comment · Reviewer_w7ox · 2026-07-20
> > > > **Official Comment**
> > > >
> > > > I appreciate the detailed reply, which resolved many of the concerns.

---

### Review · Reviewer_dG2k · 2026-07-02

**Summary Of Contributions:**

This paper introduces **NEMO** and **SciTX** as a solution for a simple problem: When an image classifier mis-predicts, can we explain why in plain language?

Authors point out two gaps. Existing methods work at the group or dataset level,= ( or just pull a sentence from a fixed library) so they can not explain a single image whose failure is not already in that library. And there is no good way to score a free-form explanation, since metrics like BLEU or ROUGE need a reference answer that does not exist here.

Their solution has two parts:

NEMO is a benchmark of 1200 misclassified images from ImageNet-R, ObjectNet, and ImageNet-D, each dataset built to vary one known *concept*. A good explanation should name that *concept*, and an LLM judge scores whether it does, a metric they call LLM Match.

SciTX is the method: for each wrong prediction, SciTX pull reference images the model gets right from both the true and predicted classes, asks a VLM to generate several candidate explanations, and then it tests each one. It turns each explanation into an **intervention**, like editing the image, or rewriting the prompt for CLIP-style models, to remove the aspect the explanation blames. It keeps whichever one moves the model back toward the right answer the most. That prediction shift is a second metric, CEI.

On NEMO, SciTX beats a retrieval baseline and two MLLM baselines on both metrics, across all three datasets and three classifiers (ViT, CLIP, SigLIP).

Authors also have conducted small scale human study to show the effectiveness of their method.

**Audience:**

Yes

**Audience Explanation:**

Yes. The problem is real and current, and a sample-level, generation-based approach to explaining classifier errors is interesting, and people working on model debugging, explainability, and VLM/MLLM-as-judge would find both the method and the benchmark worth knowing.

That said, the framing oversells the fit to the stated motivation, and I would want this made more realistic. The introduction sells the problem through *non-experts* debugging deployed models and making"trust" or "retrain decisions", but the paper never tests a debugging outcome, and the human study recruits AI practitioners rather than the non-experts it invokes.


The benchmark is also built entirely from off-the-shelf, heavily-used robustness datasets (ImageNet-R, ObjectNet, ImageNet-D), which are convenient, but far from a realistic debugging setting where the cause of an error is unknown, unlabeled, and often not a single clean factor.

So the findings are of interest, but the paper would be stronger, and more honest to its audience, if the framing matched what is actually demonstrated.

**Broader Impact Concerns:**

The paper includes a broader-impact statement and I think it is okay. I think the main risk is a gap between how convincing model generated explanations are and how correct they are. And I have no concerns about the data or the human study beyond the standard ones.

**Claims And Evidence:**

Yes

**Claims Explanation:**

I think the core empirical claims are supported.

The central claim is that *SciTX produces better per-image error explanations than the retrieval and MLLM baselines*, and the evidence for this exists:

 - Table 1 shows SciTX winning on both metrics across all three datasets and all three classifiers, so this is a broad and consistent result, not one or two cherry-picked cells
-  Table 2 addresses the obvious worry that CEI is baked into SciTX's own selection, by giving the baselines the same generate-and-select step, and SciTX still leads.
- Tables 3, 4, and 5 back the smaller claims that the ranking survives a change of image editor, improves with a stronger backbone, and benefits from the class-aware retrieval.




Separately, a few of the paper's framing claims run ahead of the evidence, around CEI's independence, what LLM Match actually measures, and the "faithfulness" language.

**Requested Changes:**

- Address the LLM Match provenance shortcut: Because LLM Match scores whether an explanation names the dataset's controlled factor, and that factor is inferable from which dataset the image came from, a fixed dataset-keyed template ("this failed due to an unusual viewpoint or background" for ObjectNet) would name the factor without ever looking at the image, and should score highly. Please add this constant baseline. If it approaches or beats the real baselines on LLM Match, that needs to be shown and discussed, and it bears directly on whether LLM Match should be called the primary metric.


- Scope the claims to what the evidence shows: First, CEI is both what SciTX optimizes and one of the two headline metrics, so it is not a fully independent test of explanation quality, and the paper effectively leans on LLM Match as the independent check. Please state this plainly in the metric section, not only in passing, so CEI is not presented as if it independently validates SciTX. Second, describing the explanations as "grounded in the model's actual failure mechanism" or "faithful" is stronger than a prediction shift, from an edit the method designed itself, can establish. Please replace this with the accurate framing.

- Demonstrate the out-of-library claim, or rescope the motivation: The paper's central motivation for generation over retrieval is that retrieval cannot describe errors whose cause is absent from its fixed corpus. This is asserted, and used to explain retrieval's low scores, but never tested as a coverage effect. Retrieval's dataset-level swing (decent on ImageNet-R, low on ObjectNet and ImageNet-D) is confounded, since the three datasets differ in difficulty and in how easily the factor can be named, not only in corpus coverage, and the corpus is never characterized, so a low score is equally consistent with weak retrieval or over-generic corpus sentences as with out-of-corpus causes. Please either (a) partition errors into in-corpus and out-of-corpus causes and show SciTX's advantage is concentrated on the out-of-corpus subset, or (b) vary corpus coverage and show retrieval degrades while SciTX stays flat.

- Distinguish"preferred" from "correct": Because faithfulness is not established, a human preference win shows the explanations are more convincing, not more correct, and a confident but wrong explanation is more harmful to a real user than a vague one. Please state this explicitly where the human results are discussed, and avoid implying that higher "actionability" or "factuality" rankings mean the explanations are causally accurate.

- Strengthen or rescope the human study:  First, the 30 samples are filtered to high harmonic-mean CEI across methods, which selects on the quantity SciTX optimizes and over-represents images where a clean intervention exists, so it is not representative of the benchmark. Please report results without this filter, or justify it explicitly and note the limitation. Second, the study uses AI practitioners, not the non-experts in the motivation, so please either recruit representative users or narrow the framing.

---

> ### Author Response · Authors · 2026-07-15
> **Response to Reviewer dG2k**
>
> We appreciate the reviewer’s careful and constructive feedback. Below, we address each point; all revisions in the manuscript are highlighted in blue.
>
> **Q1. Add a constant, dataset-keyed baseline for LLM Match**
>
> | ImageNet-R / CLIP                | LLM Match | CEI      |
> | -------------------------------- | --------- | -------- |
> | Constant Baseline                | 0.809     | 4.04     |
> | Retrieval                        | 0.270     | 11.22    |
> | CamMLLM                          | 0.066     | 3.17     |
> | ChangeMLLM                       | 0.309     | 9.59     |
> | SciTX                            | 0.632     | 17.52    |
>
> **Table. Constant baseline result.**
>
>
> We used a fixed, dataset-keyed template that specifies the controlled factor.
>
> - **High LLM Match, but a low CEI.** As indicated in the table, it achieves the highest LLM Match score among all methods (0.809, compared to SciTX’s 0.632), but performs poorly on CEI (4.04, compared to SciTX’s 17.52).
> - **Need for sample-wise explanation.** Because the template lacks sample-specific content, the resulting intervention remains generic and does not address the causes of specific failures. Merely naming a dataset’s controlled factor does not explain why a particular image is misclassified. Therefore, we jointly report LLM Match and CEI.
> - **Discussion of primary metric.** We agree that the wording should reflect this distinction: we no longer present LLM Match as the primary metric; instead, it is reported alongside CEI as a complementary signal that requires joint interpretation.
>
> We have added the constant-baseline experiment and its discussion in Appendix C.1, and revised Section 3.4 accordingly.
>
> **Q2. State that CEI does not independently validate SciTX, and remove the “faithful” and "failure mechanism" claims.**
>
> - **CEI does not independently validate SciTX.** Section 3.3 now concludes with a dedicated paragraph, “Relation to SciTX,” clarifying that SciTX uses CEI as its hypothesis-selection signal; therefore, CEI does not independently validate SciTX. Table 2 presents the effect of including or excluding this CEI-based selection for each method.
> - **Revision of wording.** We have revised the "faithful" and "failure mechanism" claims throughout the manuscript (Introduction, Related Work, and Conclusion). In particular, "grounded in the model's actual failure mechanism" (Introduction) has been removed, and "thereby evaluating its faithfulness" (Related Work) has been revised to "that measures its effect on the model's prediction."
>
> **Q3. Test the corpus-coverage claim, (a) or (b)**
>
> | ImageNet-R / CLIP | LLM Match | CEI   | Style Explanation pick |
> | ----------------- | --------- | ----- | ---------------------- |
> | Retrieval (9.3%)  | 0.242     | 11.23 | 39.5%                  |
> | Retrieval (14.0%) | 0.270     | 11.54 | 43.8%                  |
> | Retrieval (42.7%) | 0.432     | 27.58 | 77.2%                  |
> | Retrieval (57.0%) | 0.465     | 32.71 | 84.5%                  |
> | SciTX             | 0.632     | 17.60 | -                      |
>
> **Table. Ablation study of Retrieval.** The percentage in parentheses is the proportion of style sentences in the retrieval corpus. All values are re-evaluated in this experiment’s run.
>
> We did (b). We varied the number of style sentences (those matching ImageNet-R’s controlled cause) in the retrieval corpus, including a condition that enlarges the corpus with irrelevant sentences instead.
>
> - **Increased coverage improves retrieval performance.** Retrieval tracks coverage across all axes (Match 0.24 to 0.47, CEI 11 to 33, style pick 40% to 85%). SciTX remains unaffected because it does not use the corpus.
> - **57.0% condition.** We introduced style-error explanations that reference the misclassified images, which explains the high CEI; a deployed corpus would not contain such sentences in advance. Even in this scenario, retrieval’s LLM Match saturates near 0.5, earning only partial credit according to the rubric, and remains below SciTX.
> - **9.3% condition shows the opposite direction.** As relevant sentences become harder to retrieve, every axis falls below the original corpus. Retrieval therefore performs well only when the corpus already contains the causes of the errors and the retriever reliably picks them, and degrades as either condition weakens. The claim is not that SciTX wins CEI everywhere, but that SciTX is robust while retrieval is not.
>
> We have added this experiment and its discussion to Appendix C.2, and referenced it in Section 4.2.1 where the coverage claim is addressed.

---

> > ### Author Response · Authors · 2026-07-15
> >
> > **Q4. Distinguish “preferred” from “correct” in the human study.**
> >
> > We agree with this distinction and now state it explicitly.
> >
> > - **Preference does not equate to correctness (Section 4.3).** The Results paragraph of the human evaluation now clarifies that the rankings reflect perceived helpfulness rather than correctness, and that a convincing but incorrect explanation could also receive a high ranking.
> > - **Other mentions.** The Abstract, Introduction, and Conclusion report only the rankings on helpfulness dimensions, so we leave them unchanged.
> >
> >
> > **Q5. Justify the harmonic-mean filter and rescope the participant framing.**
> >
> > - **Justification of the harmonic-mean filter.** We selected the harmonic mean precisely to avoid cherry-picking samples in favor of any single method; it is dominated by the lowest-scoring method, so a sample passes only when even the weakest baseline yields a non-trivial explanation. We acknowledge the limitation noted by the reviewer: it over-represents samples whose causes are amenable to intervention. The Study Design paragraph (Section 4.3) now includes this justification.
> > - **Rescope of the participant framing.** We have narrowed the framing. Ranking error explanations on dimensions such as factuality and actionability requires sufficient machine learning background, which is why we recruited AI practitioners. Section 4.3 now states this explicitly.

---

> > > ### Comment · Reviewer_dG2k · 2026-07-20
> > >
> > > I thank the authors for their detailed response, as well as the additional experiments and revisions they have made.